# STARSS23: An Audio-Visual Dataset of Spatial Recordings of Real Scenes with Spatiotemporal Annotations of Sound Events

**Kazuki Shimada**
Sony AI
`kazuki.shimada@sony.com`

**Archontis Politis**
Tampere University
`archontis.politis@tuni.fi`

**Parthasaarathy Sudarsanam**
Tampere University

**Daniel Krause**
Tampere University

**Kengo Uchida**
Sony AI

**Sharath Adavanne**
Tampere University

**Aapo Hakala**
Tampere University

**Yuichiro Koyama**
Sony Group Corporation

**Naoya Takahashi**
Sony AI

**Shusuke Takahashi**
Sony Group Corporation

**Tuomas Virtanen**
Tampere University

**Yuki Mitsufuji**
Sony AI, Sony Group Corporation

## Abstract

While direction of arrival (DOA) of sound events is generally estimated from multichannel audio data recorded in a microphone array, sound events usually derive from visually perceptible source objects, e.g., sounds of footsteps come from the feet of a walker. This paper proposes an audio-visual sound event localization and detection (SELD) task, which uses multichannel audio and video information to estimate the temporal activation and DOA of target sound events. Audio-visual SELD systems can detect and localize sound events using signals from a microphone array and audio-visual correspondence. We also introduce an audio-visual dataset, Sony-TAu Realistic Spatial Soundscapes 2023 (STARSS23), which consists of multichannel audio data recorded with a microphone array, video data, and spatiotemporal annotation of sound events. Sound scenes in STARSS23 are recorded with instructions, which guide recording participants to ensure adequate activity and occurrences of sound events. STARSS23 also serves human-annotated temporal activation labels and human-confirmed DOA labels, which are based on tracking results of a motion capture system. Our benchmark results demonstrate the benefits of using visual object positions in audio-visual SELD tasks. The data is available at `https://zenodo.org/record/7880637`.

## 1 Introduction

Given multichannel audio input from a microphone array, a sound event localization and detection (SELD) system [1, 45, 8, 46] outputs a temporal activation track for each of the target sound classes along with one or more corresponding spatial trajectories, e.g., the direction of arrival (DOA) around the microphone array, when the track indicates activity. Such a spatiotemporal characterization of sound scenes can be used in a wide range of machine cognition tasks, such as inference on the type of environment, tracking of specific types of sound sources, acoustic monitoring, scene visualization systems, and smart-home applications. Recently neural network (NN)-based SELD

37th Conference on Neural Information Processing Systems (NeurIPS 2023) Track on Datasets and Benchmarks.

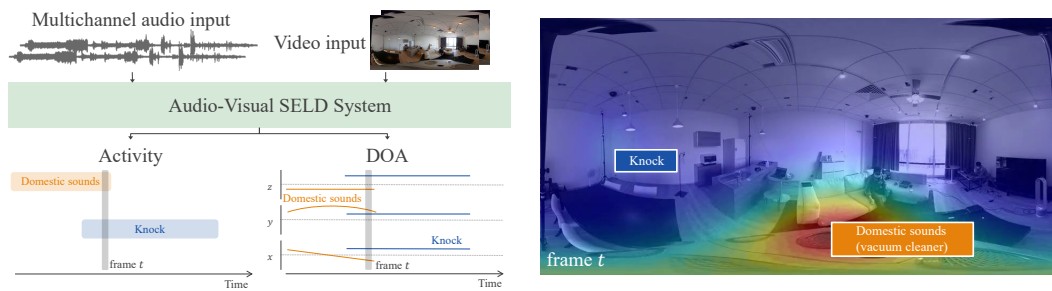

Figure 1: An audio-visual SELD system takes multichannel audio recording data and video data aligned with the audio data and estimates temporal activity and DOA of each class per frame. In a frame $t$ of a recording in STARSS23, a knocking sound comes from a door while a domestic sound derives from a vacuum cleaner behind a microphone array, as shown in an overlay of 360° video and spatial acoustic power map.

systems [1, 45, 8, 46] show high localization and detection performance. These systems need data with activity and DOA labels of target sound events for training and evaluation. Because annotation of DOA labels is challenging in real sound scene recordings, most SELD datasets [2, 36, 35, 19, 28] consist of synthetic audio data, which are made by convolution of monaural sound event signals and multichannel impulse response signals with DOA labels. The Sony-TAu Realistic Spatial Sound-scapes 2022 dataset (STARSS22) [39] tackles real sound scene recordings with DOA labels, which is based on tracking results of a motion capture (mocap) system. Currently, STARSS22 is the only SELD dataset with real sound scenes, including overlapping sound events, moving source events, and natural distribution of temporal activation and DOA, e.g., sounds of footsteps are relatively short and heard from a lower elevation. While this dataset is suitable for evaluating audio-only SELD systems in natural sound scenes, the dataset does not include other modality input, e.g., video data.

Sound events in real sound scenes originate from their sound source objects, e.g., speech comes from a person's mouth, sounds of footsteps are produced from the feet of a walker, and a knocking sound originates from a door. Such sound source object information is usually apparent in the visual modality. Video data aligned with audio recordings in SELD tasks have the potential to mitigate difficulties and ambiguities of the spatiotemporal characterization of the sound scene as audio-visual data improves source separation [59, 14, 58, 26] and speech recognition [32, 3, 34, 24]. Visible people in the video data can provide candidate positions of human body-related sounds. When a person walks in the video, tapping sounds are easily recognized as footsteps. To investigate how a visual feature input affects a SELD system, we propose an *audio-visual SELD* task, which uses audio and video data to estimate spatiotemporal characterization of sound events. The left side of Figure 1 shows an audio-visual SELD system, which takes multichannel audio recordings and video aligned with the audio recordings and outputs activity and DOA of target sound events in each frame. To tackle audio-visual SELD tasks, we need an audio-visual dataset consisting of multichannel audio, video, and activity and DOA labels of sound events per frame.

There is another interest in audio-visual sound source localization tasks [44, 12, 61, 4, 33], which takes monaural audio and images and estimates where the audio is coming from in the images. They focus on learning audio-visual *semantic* correspondence, not estimating *physical* DOA around a microphone array. While the audio-visual sound source localization datasets [44, 12, 61] are adequate to train NNs with audio-visual correspondence and evaluate localization performance in images, they are typically monophonic without spatial labels for the target sound events. Several datasets serve multichannel audio and video data with real sound scenes [27, 53, 54, 60, 56, 10, 40, 41, 50], whereas only a few audio-visual datasets have multichannel audio data with DOA labels of speakers around a microphone array [40, 41, 50]. While these datasets help to evaluate audio-visual speaker DOA estimation (DOAE) tasks, the evaluation focuses only on speech, not sound events such as musical instruments or footsteps.

To tackle audio-visual SELD tasks, we introduce an audio-visual dataset, the *Sony-TAu Realistic Spatial Soundscapes 2023 (STARSS23)*, consisting of multichannel audio, video, and spatiotemporal annotations of sound events, i.e., activity and DOA labels per each frame. The right part of Figure 1 shows a still in a frame of a video in STARSS23, made from 360° video, spatial acoustic power

Table 1: We compare STARSS23 to other real sound scene datasets from three points of view: whether audio data is multichannel, whether video data is included, and what annotation type is.

| Dataset | Multichannel audio | Video | Annotation |
|---|---|---|---|
| STARSS22 [39] | ✓ | - | Activity & DOA of sound events |
| VGG-SS [12] | - | ✓ | Bounding box |
| AVSBench [61] | - | ✓ | Segmentation map |
| YouTube-360 [27] | ✓ | ✓ | - |
| AVRI [41] | ✓ | ✓ | Activity & DOA of speech |
| STARSS23 (Ours) | ✓ | ✓ | Activity, DOA & distance of sound events |

map generated from a microphone array, and sound event labels. The dataset contains over 7-hour recordings with 57 participants in 16 rooms with the spatiotemporal annotation as a development set. The participants are guided by generic instructions and suggested activities in the recording to induce adequate occurrences of the sound events and diversity of content. There are 13 classes of target sound events, such as speech, musical instruments, and footsteps. To reveal whether the dataset has a natural distribution of sound events, we analyze the dataset regarding frame coverage, overlap, and DOA distributions per each sound event class. We develop and test an audio-visual SELD system with STARSS23. To investigate the effects of audio-visual input, we present overall localization and detection performance and per-class results.

## 2 Related Work

**Sound event localization and detection** SELDnet [1] is the first SELD method, which uses convolutional recurrent neural network (CRNN) to output activity and DOA separately. An activity-coupled Cartesian DOA (ACCDOA) [45] vector, which embeds sound event activity information to the length of a Cartesian DOA vector, enables us to solve SELD tasks with a single output. While the two methods tackle overlaps from different classes, they cannot solve overlaps from the same class. Multi-ACCDOA [46] is an ACCDOA extension, allowing models to output overlaps from the same class. To handle that case effectively, Multi-ACCDOA incorporates auxiliary duplicating permutation invariant training (ADPIT) [46]. There are other SELD works about framework [8, 29, 55], network architecture [48], audio feature [30, 21], and data augmentation [51, 42].

To train or evaluate SELD methods, we need multichannel audio data with temporal activation and DOA labels. In synthetic multichannel audio datasets [2, 36, 35, 19, 28], DOA labels can be easily annotated because the data are made from multichannel impulse response signals with DOA labels. While the SECL-UMons and AVECL-UMons datasets [6, 5] tackled spatial recording with DOA labels, it is limited to isolated single event recordings or combinations of two simultaneous events, ignoring spatiotemporal information linking events in a natural scene. STARSS22 [39] tackled real spatial recording with temporal activation and DOA labels of each target class in natural scenes. Participants improvise natural scenes with a mocap system, whose tracking results are used for DOA labels. However, the dataset does not release video data. Therefore, it cannot be used to evaluate audio-visual SELD systems.

**Audio-visual sound source localization** There is broad interest in audio-visual sound source localization tasks [44, 12, 61, 4, 33]. Chen *et al.* have tackled unsupervised learning to localize sound sources in video and evaluated the method on the VGG-SS dataset [12], which annotates bounding boxes of sound sources for sound and video pairs. The AVSBench dataset [61] serves pixel-level audio-visual segmentation maps for videos over 23 class categories. Because the datasets do not have multichannel audio recordings, they cannot be applied to evaluating SELD tasks.

**Audio-visual dataset with multichannel audio** Several audio-visual datasets include multichannel audio data [27, 53, 54, 60, 56, 10, 40, 41, 50]. As many datasets are used for self-supervised learning [27, 54, 53] or non-localization tasks [60, 56, 10], there are no DOA labels. The YouTube-360 dataset [27] serves first-order ambisonics (FOA) signal and 360° video data without any labels for self-supervised learning.

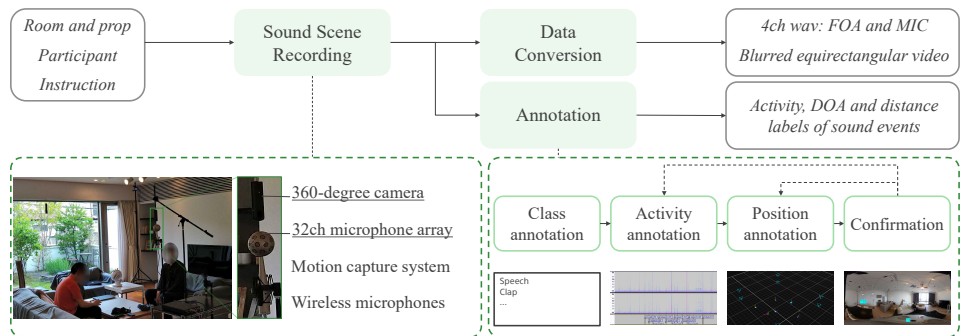

Figure 2: The pipeline of data construction.

A few audio-visual datasets are collected for audio-visual DOAE tasks [40, 41, 50]. Qian *et al.* proposed an audio-visual DOAE system, which takes spectrograms and phase features from the audio input and face-bounding boxes from video input to estimate the DOA of each speaker [41]. The system was evaluated with the Audio-Visual Robotic Interface (AVRI) dataset, recorded using Kinect and a four-channel Respeaker array, along with activity and DOA labels. The audio-visual features are helpful for DOAE, and the dataset supports the evaluation of audio-visual speaker DOAE. However, the dataset is only for speech, not various sound events such as clapping and knocks. We summarize the comparison of STARSS23 with other real sound scene datasets in Table 1.

## 3  STARSS23 Dataset

### 3.1  Overview

STARSS23 contains multichannel audio and video recordings of sound scenes in various rooms and environments, together with temporal and spatial annotations of sound events belonging to a set of target classes. The dataset enables us to train and evaluate audio-visual SELD systems, which localize and detect sound events from multichannel audio and visual information. STARSS23 is available in a public research data repository[1] under the MIT license. There is also a demo video[2].

The contents are recorded with short instructions, guiding participants in improvising sound scenes. The recordings contain a total of 13 target sound event classes. The multichannel audio data are delivered as two 4-channel spatial formats: FOA and tetrahedral microphone array (MIC). The video data are anonymized 1920×960 equirectangular videos recorded by a 360° camera. The annotations of STARSS23 consist of temporal activation, DOA, and source distance of the target classes. STARSS23 is split into a development set and an evaluation set. The development set totals about 7 hours and 22 minutes, of which 168 clips were recorded with 57 participants in 16 rooms. The development set is further split into a training part (dev-set-train, 90 clips) and a testing part (dev-set-test, 78 clips) to support the development process. In the evaluation set, no publicly available annotations exist because the evaluation set is prepared for a competition, which is described in Appendix D.

STARSS23 improves a multichannel audio dataset, i.e., STARSS22 [39]. One of the critical differences is releasing video data aligned with multichannel audio data. While we maintain all the sessions of STARSS22, we add about 2.5 hours of material to the development set. STARSS23 also serves source distance labels of sound events as additional annotations. We follow the recording procedure in STARSS22, where video data are used only to check labels internally. Adding descriptions about video data and distance annotation, we show the data construction part as an audio-visual dataset.

### 3.2  Data Construction

As shown in Figure 2, STARSS23 is constructed in three steps: sound scene recording, data conversion, and annotation. We explain each step as follows.

---

[1] https://zenodo.org/record/7880637
[2] https://www.youtube.com/watch?v=ZtL-8wBYPow

**Sound scene recording**  STARSS23 was created in Tampere, Finland, and Tokyo, Japan. Recordings at both sites shared the same process, organized in sessions corresponding to different rooms, sound-making props, and participants. In each session, various clips were recorded with combinations of that session's participants acting simple scenes and interacting among themselves and with the sound props. The scenes were based on generic instructions on the desired sound events. The instructions were a rough guide to ensure adequate event activity and occurrences of the target sound classes in a clip. The left photo of Figure 2 shows that participants improvise following the instructions.

A set of 13 target sound event classes are selected to be annotated, based on the sound events captured adequately in the recorded scenes. The class labels are chosen to conform to the AudioSet ontology [17]. They are: ***female speech, male speech, clapping, telephone, laughter, domestic sounds, footsteps, door, music, musical instrument, water tap, bell, knock***. Music, e.g., background music or pop music, is played by a loudspeaker in the room. On the other hand, musical instruments are played by participants, including acoustic guitar, piano, and others. Domestic sounds consist of vacuum cleaners, mechanical fans, and boiling, which have strongly directional and loud sounds. They can be distinguishable from natural background noise in sound scenes. The scenes also contain directional interference sounds, such as computer keyboard or shuffling cards that are not labeled.

As shown in the left photos of Figure 2, each scene was captured with audio-visual sensors, i.e., a high-resolution 32-channel spherical microphone array (Eigenmike em32[3]) with a height set at 1.5 m, and a 360° camera (Ricoh Theta V[4]) mounted 10 cm above the microphone array. For each recording session, a suitable position of the Eigenmike and Ricoh Theta V was determined to cover the scene from a central place. We also captured the scenes with two additional sensors for annotation: a mocap system of infrared cameras surrounding the scene, tracking reflective markers mounted on the participants and sound sources of interest (Optitrack Flex 13[5]), and wireless microphones mounted on the participants and sound sources, providing close-mic recordings of the main sound events (Røde Wireless Go II[6]). The origin of the mocap system was set at ground level on the exact position of the Eigenmike. In contrast, the mocap cameras were positioned at the room's corners.

Recording starts on all devices before the beginning of a scene and stops right after. A clapper sound initiated the acting, and it served as a reference signal for synchronization between the different types of recordings, including the mocap system that can record a monophonic audio side signal for synchronization. All types of recordings were manually synchronized based on the clapper sound and subsequently cropped and stored at the end of each recording session. The details of sound scene recording, e.g., generic instructions, sound events, and sensors, are summarized in Appendix A.1.

**Data conversion**  The original 32-channel recordings were converted to two 4-channel spatial formats: FOA and MIC. Conversion of the Eigenmike recordings to FOA following the SN3D normalization scheme (or ambiX) was performed with measurement-based filters [37]. Regarding the MIC format, channels 6, 10, 26, and 22 of the Eigenmike were selected, corresponding to a nearly tetrahedral arrangement. Analytical expressions of the directional responses of each format can be found in [36]. Finally, the converted recordings were downsampled to 24kHz. The raw 360° video data were converted to an equirectangular format with 3840×1920 resolution at 29.97 frames per second, which was convenient to handle as planar video data. Based on the participant's consent, the visible faces of all recordings were blurred. Finally, the video with face-blurring was converted to a 1920×960 resolution.

**Annotation**  Spatiotemporal annotations of the sound events were conducted manually by the authors and research assistants. As shown in the lower right part of Figure 2, there were four steps: a) annotate the subset of the target classes that were active in each scene, b) annotate the temporal activity of such class instances, c) annotate the position of each such instance when active, moreover, d) confirm the annotations. Class annotations (a) were observed and logged during each scene recording. Activity labels (b) were manually annotated by listening to the wireless microphone recordings. Because each wireless microphone would capture prominent sounds produced by the participant or source it was assigned to, onset, offsets, source, and class information of each event could be conveniently extracted. In scenes or instances where associating an event to a source was

---

[3]https://mhacoustics.com/products#eigenmike1

[4]https://theta360.com/en/about/theta/v.html

[5]https://optitrack.com/cameras/flex-13/

[6]https://rode.com/en/microphones/wireless/wirelessgoii

Table 2: Frame coverage and max, mean, and the degree of polyphony globally and of each class separately on the dev-set-train of STARSS23. The mean polyphony is computed over active frames.

| | Global | Fem. speech | Male speech | Clap | Phone | Laugh | Dom. sounds | Footsteps | Door | Music | Music. instr. | Faucet | Bell | Knock |
|---|---|---|---|---|---|---|---|---|---|---|---|---|---|---|
| Frame coverage (% total frames) | 82.5 | 28.4 | 31.4 | 0.53 | 1.02 | 2.89 | 18.6 | 2.17 | 0.70 | 22.2 | 2.32 | 0.58 | 0.99 | 0.06 |
| Max. polyphony | 6 | 2 | 3 | 2 | 1 | 4 | 2 | 3 | 1 | 2 | 2 | 1 | 1 | 1 |
| Mean polyphony | 1.47 | 1.05 | 1.06 | 1.06 | 1.00 | 1.25 | 1.03 | 1.02 | 1.00 | 1.20 | 1.28 | 1.00 | 1.00 | 1.00 |
| Polyphony 1 (% active frames) | 62.878 | 95.44 | 94.35 | 93.74 | 100 | 78.67 | 97.27 | 97.87 | 100 | 79.99 | 71.66 | 100 | 100 | 100 |
| Polyphony 2 | 28.443 | 4.56 | 5.51 | 6.26 | 0 | 18.48 | 2.73 | 1.94 | 0 | 20.01 | 28.36 | 0 | 0 | 0 |
| Polyphony 3 | 7.385 | 0 | 0.14 | 0 | 0 | 2.39 | 0 | 0.19 | 0 | 0 | 0 | 0 | 0 | 0 |
| Polyphony 4 | 1.129 | 0 | 0 | 0 | 0 | 0.46 | 0 | 0 | 0 | 0 | 0 | 0 | 0 | 0 |
| Polyphony 5 | 0.162 | 0 | 0 | 0 | 0 | 0 | 0 | 0 | 0 | 0 | 0 | 0 | 0 | 0 |
| Polyphony 6 | 0.003 | 0 | 0 | 0 | 0 | 0 | 0 | 0 | 0 | 0 | 0 | 0 | 0 | 0 |

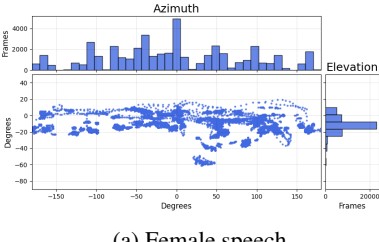

(a) Female speech

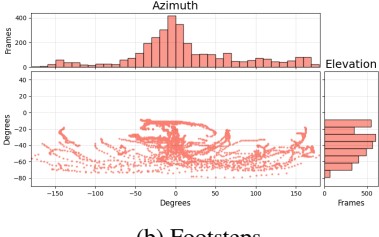

(b) Footsteps

Figure 3: The distribution of the azimuth and elevation. Each point in the scatter figure represents a frame. The top and right figures are stacked histograms of azimuth and elevation.

ambiguous purely by listening, annotators would consult the video recordings to establish the correct association. The temporal annotation resolution was set to 100 msec. After onset, offset, and class information of events was established for each source and participant in the scene, the positional annotations (c) were extracted for each such event by attaching tracking results to the temporal activity window of the event. Positional information was logged in Cartesian coordinates with respect to the mocap system's origin. The event positions were converted to spherical coordinates, i.e., azimuth, elevation, and distance, which are more convenient for SELD tasks. Then, the class, temporal, and spatial annotations were combined and converted to the text format used in the previous dataset [39]. The annotation details are summarized in Appendix A.2. Confirmation of the annotations (d) was performed by listening to the Eigenmike recording while watching a synthetic video. The video was the equirectangular one overlapped with the event activities, which were visualized as labeled markers positioned at their respective azimuth and elevation on the video plane. If a clip was not passed the confirmation, the clip was annotated again.

## 3.3 Data Analysis

Having a natural distribution of sound events is beneficial to evaluate audio-visual SELD systems. We analyze the frame coverage, polyphony, and DOA per sound event classes on the dev-set-train of STARSS23. Table 2 shows frame coverage and max, mean, and distribution of polyphony globally and of each class separately. Regular classes in frames are female and male speech, music, and domestic sounds. These classes are also frequent in our daily lives. Musical instruments and laugh classes show a high mean polyphony of the same class, which are natural situations in jam sessions and conversations. Figure 3 shows the distribution of DOA with the axis of the azimuth and elevation. Regarding female speech in Figure 3a, the elevation distribution has a strong peak around -10 degrees, while that of azimuth seems uniformly distributed. Compared to the speech class, footsteps appear in lower azimuth than -10 degrees. See Appendix B for further data analysis, e.g., duration.

## 4 Benchmark

In this section, we examine an audio-visual SELD task with STARSS23. For evaluation, we set the dev-set-train for training and hold out the dev-set-test for validation.

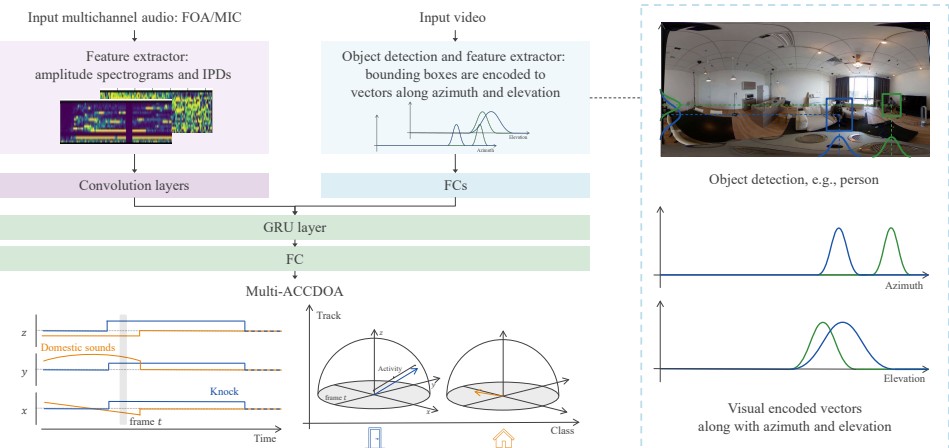

Figure 4: An audio-visual SELD system uses a CRNN model with a multi-ACCDOA output. While multi-ACCDOA output can have several tracks, we show that with a single track for simplicity. We use an object detector to incorporate visual information, which outputs bounding boxes on target classes. The results are encoded to vectors along with azimuth and elevation. Then, the encoded vectors are embedded and concatenated before the GRU layer.

## 4.1 Audio-Visual SELD System

To build an audio-visual SELD system, we start with an audio-only system based on SELDnet [1] and multi-ACCDOA [46], which is widely used in audio-only SELD tasks [39, 43, 31]. To extend the audio-only systems to audio-visual, we merge visual and audio information in the middle of the network, following the audio-visual speaker DOAE work [41].

First, we summarize the audio-only SELD system. Let $\boldsymbol{X}_\mathrm{a} \in \mathbb{R}^{M \times F \times T}$ be an input consisting of $M$-channel $F$-dimensional $T$-frame audio features, e.g., amplitude spectrograms. Convolutional layers are used as an audio embedding network, $\mathcal{F}_{\mathrm{a},\theta_\mathrm{a}}(\cdot)$, where $\theta_\mathrm{a}$ is the network parameters:

$$\boldsymbol{E}_\mathrm{a} = \mathcal{F}_{\mathrm{a},\theta_\mathrm{a}}(\boldsymbol{X}_\mathrm{a}), \tag{1}$$

where $\boldsymbol{E}_\mathrm{a} \in \mathbb{R}^{K_\mathrm{a} \times T'}$ is an audio embedding. $K_\mathrm{a}$ and $T'$ are the length of an embedding vector and the number of time frames, respectively. The embedding is followed by an audio-only decoder network with parameter $\theta_\mathrm{ao}$, $\mathcal{F}_{\mathrm{ao},\theta_\mathrm{ao}}(\cdot)$, which consists of a gated recurrent unit (GRU) layer and a fully connected layer (FC):

$$\hat{\boldsymbol{P}} = \mathcal{F}_{\mathrm{ao},\theta_\mathrm{ao}}(\boldsymbol{E}_\mathrm{a}), \tag{2}$$

where $\hat{\boldsymbol{P}} \in \mathbb{R}^{3 \times N \times C \times T''}$ is an estimated output of $N$-track $C$-class $T''$-frame multi-ACCDOA. Each class of the multi-ACCDOA output in track $n$ and frame $t$ is represented by a three-dimensional vector with Cartesian coordinates $x$, $y$, and $z$, as shown in the bottom of Figure 4. The length of the vector shows activity, and the direction of the vector indicates the DOA around the microphone array. To train the SELD system, we use mean squared error (MSE) between the estimated and target multi-ACCDOA outputs under the ADPIT scheme [46]. In inference, when the length of the vector is greater than a threshold, the class is considered active.

Then, we prepare a visual embedding. As visual input, we use the corresponding video frame at the start of the audio features. With the corresponding image, an object detection module, e.g., YOLOX [15], outputs bounding boxes of potential objects on target classes, e.g., person class. Let us denote a bounding box as $(u_b, v_b, w_b, h_b)$, where $u_b$ and $v_b$ indicate the image position of the top-left corner, and $w_b$ and $h_b$ show the width and height of the bounding box. Each bounding box is encoded to two vectors $\rho_\mathrm{azi}(u)$ and $\rho_\mathrm{ele}(v)$ along the image's horizontal axis $u$ and vertical axis $v$. We follow the same formulation in [41] using a Gaussian distribution:

$$\rho_\mathrm{azi}(u) = \exp\left(\frac{-|u - \mu_u|^2}{\sigma_u^2}\right), \tag{3}$$

where the center $\mu_u = u_b + \frac{1}{2}w_b$ is the horizontal center of the bounding box, and the standard deviation $\sigma_u$ is proportional to $w_b$. The vertical vector $\rho_{\text{ele}}(v)$ has the same formula as $\rho_{\text{azi}}(u)$ by replacing $u$ with $v$, $\mu_u$ with $\mu_v = v_b + \frac{1}{2}h_b$, and $\sigma_u$ with $\sigma_v$ in Equation 3. These vectors are concatenated into a visual feature input $\boldsymbol{X}_{\text{v}} \in \mathbb{R}^{2 \times B \times L}$. $B$ is the maximum number of bounding boxes, and $L$ is the length of the vectors. When no bounding box is detected, the visual feature input is set to an all-zero tensor. The right part of Figure 4 depicts the visual feature input. FCs are used as a visual embedding network, $\mathcal{F}_{\text{v},\theta_{\text{v}}}(\cdot)$, where $\theta_{\text{v}}$ is the network parameters:

$$\boldsymbol{E}_{\text{v}} = \mathcal{F}_{\text{v},\theta_{\text{v}}}(\boldsymbol{X}_{\text{v}}), \tag{4}$$

where $\boldsymbol{E}_{\text{v}} \in \mathbb{R}^{K_{\text{v}}}$ is a $K_{\text{v}}$-dimensional visual embedding vector.

Using the visual embedding vector, we finally extend the audio-only system into an audio-visual one. The visual embedding vector is repeated $T'$ times to align the both embeddings in the time axis. Then the repeated visual embedding $\boldsymbol{E}'_{\text{v}} \in \mathbb{R}^{K_{\text{v}} \times T'}$ and the audio embedding are concatenated and fed into an audio-visual decoder network with parameter $\theta_{\text{av}}$, $\mathcal{F}_{\text{av},\theta_{\text{av}}}(\cdot)$, to get multi-ACCDOA output:

$$\hat{\boldsymbol{P}} = \mathcal{F}_{\text{av},\theta_{\text{av}}}(\boldsymbol{E}_{\text{a}}, \boldsymbol{E}'_{\text{v}}). \tag{5}$$

## 4.2 Evaluation Metric

We used four joint localization and detection metrics [25, 38], which are widely-used in audio-only SELD tasks [46, 39, 31]. Two metrics are referred to as location-aware detection and are error rate ($ER_{20°}$) and F-score ($F_{20°}$) in one-sec non-overlapping segments. We consider a prediction as a true positive if the prediction and reference class are the same and the angle difference is below $20°$. $F_{20°}$ is calculated from location-aware precision and recall, whereas $ER_{20°}$ is the sum of insertion, deletion, and substitution errors, divided by the total number of the references. The other two metrics are referred to as class-aware localization and are localization error ($LE_{CD}$) in degrees and localization recall ($LR_{CD}$) in one-sec non-overlapping segments, where the subscript refers to classification-dependent. Unlike location-aware detection, we do not use any threshold but estimate the difference between the correct prediction and reference. $LE_{CD}$ expresses the average angular difference between the same class's predictions and references. $LR_{CD}$ tells the true positive rate of how many of these localization estimates were detected in a class out of the total number of class instances. We used the macro mode of computation while the mode does not apply $ER_{20°}$ because it includes substitution errors between two classes. We first computed the metrics for each class and then averaged them for the other three metrics to obtain the final system performance. See Appendix C.1 for further details.

## 4.3 Experimental Setting

As audio features, multichannel amplitude spectrograms and inter-channel phase differences (IPDs) are used [46]. Input features are segmented to have a fixed length of 1.27 sec. To reduce the calculation cost of video, we use $360 \times 180$ videos converted from the released $1920 \times 960$ videos. As visual input, we extract the corresponding video frame at the start of the audio features. We use a pretrained YOLOX object detection model[7] to get bounding boxes of person class. Other classes, e.g., cell phone or sink, are not stably detected in our preliminary experiments with STARSS23 videos. The bounding box results are encoded to vectors along azimuth and elevation as in Sec.4.1. The maximum number of bounding boxes is $B = 6$. The vector size is $L = 37 (= 36 + 1)$ to cover 360 degrees of azimuth per 10 degrees. While we tested convolutional neural network (CNN)-based visual feature extraction methods [47, 20], they performed worse than the object detection method.

To get audio embedding, we stack three convolutional layers with kernel size $3 \times 3$. We embed the visual encoded vectors with two FCs. The lengths of both embedding vectors are $K_{\text{a}} = K_{\text{v}} = 64$. The concatenated embeddings are processed with a bidirectional GRU layer with a hidden state size of 256. The number of target classes was $C = 13$. The number of tracks in the multi-ACCDOA format was fixed at $N = 3$ maximum simultaneous sources. The threshold for activity was 0.3 to binarize predictions during inference.

The experiments are for the two formats: FOA and MIC. We compare the audio-visual SELD system with an audio-only system based on the same data split and implementation. The difference is the

---

[7]https://github.com/open-mmlab/mmdetection/blob/master/configs/yolox/yolox_tiny_8x8_300e_coco.py

Table 3: SELD performance ($C = 13$) in audio-visual and audio-only systems evaluated for the dev-set-test in STASS23.

| Format | System | $ER_{20°} \downarrow$ | $F_{20°} \uparrow$ | $LE_{CD} \downarrow$ | $LR_{CD} \uparrow$ |
|--------|--------|-----------|-----------|-----------|-----------|
| FOA | Audio-Visual | $1.03 \pm 0.03$ | $13.2 \pm 0.6$ % | $51.6 \pm 5.6$ ° | $34.2 \pm 1.0$ % |
|     | Audio-Only | $0.97 \pm 0.05$ | $14.8 \pm 0.3$ % | $55.9 \pm 4.0$ ° | $35.1 \pm 2.8$ % |
| MIC | Audio-Visual | $1.08 \pm 0.04$ | $10.0 \pm 0.8$ % | $60.0 \pm 3.3$ ° | $27.6 \pm 1.7$ % |
|     | Audio-Only | $1.04 \pm 0.02$ | $10.7 \pm 2.1$ % | $66.9 \pm 6.8$ ° | $29.6 \pm 3.0$ % |

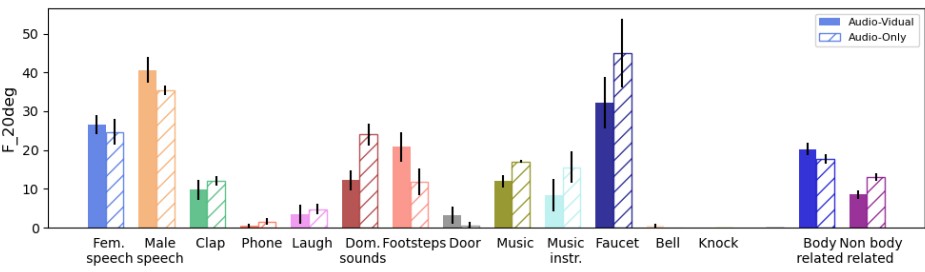

Figure 5: Location-aware F-score per sound event class ($C = 13$) in audio-visual and audio-only systems evaluated for the dev-set-test in STASS23.

presence or absence of video input. Because the visual input is bounding boxes of person class, we especially focus on five classes related to the human body, i.e., female and male speeches, clapping, laughing, and footsteps. To further investigate the effect of the visual input on the body-related classes, we add experiments where we set only the body-related classes as the target classes of the SELD systems. The systems are trained and evaluated with activities and DOAs of speeches, clap, laugh, and footsteps classes, i.e., $C = 5$. The other settings are the same as the 13-class SELD experiments. Details on the experimental setting are in Appendix C.2. The code of training, inference, and evaluation is available in a GitHub repository[8] under the MIT license.

## 4.4 Experimental Results

Table 3 summarizes the 13-class SELD performance of the audio-visual and audio-only SELD systems in both audio formats. Compared with both formats, SELD systems with FOA format show better SELD performance than those with MIC format. In FOA format, while the audio-visual SELD system shows a slightly worse location-aware F-score, the audio-visual system exhibited better localization error with comparable localization recall. There is a similar trend of lower localization error in MIC format.

We investigate the location-aware F-score over 13 classes, considering both localization and detection aspects. Figure 5 shows the F-score per class in FOA format. We show the average score of the five body-related classes on the left of the figure. The audio-visual system demonstrates a higher location-aware F-score in the body-related. On the other hand, the audio-visual system performs worse in the average of the other classes, i.e., non-body-related. The results suggest that the visual input, i.e., bounding boxes of a person, contributes to localization and detection of body-related classes, whereas the visual input might limit the performance of non-body-related classes.

Table 4 shows the body-related classes only SELD performance in audio-visual and audio-only systems. The audio-visual SELD system outperforms the audio-only system in all metrics. The visual location information of people enables the audio-visual system to localize and detect body-related classes more accurately. Audio-only system in MIC format shows high standard deviation in localization error. It is because a few classes are sometimes set $180°$ as they had no true positive output. Even if we omit such cases, the audio-visual system still shows lower localization error.

---

[8] https://github.com/sony/audio-visual-seld-dcase2023

Table 4: Body-related classes only SELD performance ($C = 5$) in audio-visual and audio-only systems evaluated for the dev-set-test in STASS23.

| Format | System | $ER_{20°}$ ↓ | $F_{20°}$ ↑ | $LE_{CD}$ ↓ | $LR_{CD}$ ↑ |
|--------|--------|--------------|-------------|-------------|-------------|
| FOA | Audio-Visual | $0.93 \pm 0.08$ | $22.4 \pm 2.4\,\%$ | $31.0 \pm 3.6\,°$ | $45.0 \pm 5.0\,\%$ |
|  | Audio-Only | $0.93 \pm 0.02$ | $18.0 \pm 1.9\,\%$ | $33.8 \pm 0.8\,°$ | $40.9 \pm 5.5\,\%$ |
| MIC | Audio-Visual | $0.95 \pm 0.01$ | $20.9 \pm 1.9\,\%$ | $28.9 \pm 1.5\,°$ | $37.4 \pm 4.6\,\%$ |
|  | Audio-Only | $0.97 \pm 0.10$ | $16.0 \pm 3.7\,\%$ | $58.4 \pm 26.8\,°$ | $30.9 \pm 10.9\,\%$ |

Our experimental results show that visual location information of people improves SELD performance in human body-related classes. If visual location information of people is provided, we should expect improvements in the human body-related classes, while we should not expect improvements in the rest. The results suggest that if visual information matches the target classes of an audio-visual SELD system, the audio-visual system performs better.

## 5 Discussion on Limitations and Future Work

Like any visual machine learning system, deploying audio-visual SELD systems might lead to privacy issues, e.g., capturing raw visual information, if the consent of the people being recorded is not considered. We also recognize that improved SELD capability could empower surveillance technology without appropriate regulation.

As shown in Table 2, STARSS23 is imbalanced and long-tailed. When considering it as an evaluation set, the dataset would be appropriate as it can reflect real soundscape, which is usually imbalanced and long-tailed. On the other hand, if we use it as a training set, the imbalance and long-tail make it difficult for a model to learn about rare categories. We can use machine learning algorithms [22, 7] or data sampling strategies [9] to tackle the data imbalance problem.

The amount of STARSS23 is relatively small as training data. We can use a multichannel audio-visual simulation platform, e.g., SoundSpaces 2.0 [11], for geometry-based audio rendering in 3D environments. Also, we can use multichannel audio-only data such as TAU-NIGENS Spatial Sound Events (TNSSE) datasets [2, 36, 35] to train an audio branch separately. Unlabeled multichannel audio-visual data, e.g., YouTube-360 [27], could help the pre-training for audio-visual SELD tasks.

There is room for developing visual features and model structure to improve the audio-visual SELD performance. While the visual location information of people improves the performance in the body-related classes, we need to develop practical visual features for the non-body-related classes. A more sophisticated model structure than the current concatenation could boost the performance. A recent work made the audio-visual SELD system better than the audio-only one in STARSS23, using decision-level audio-visual fusion with keypoints, e.g., feet or telephones [52].

While we apply STARSS23 for audio-visual SELD tasks, STARSS23 would be applied to other tasks, e.g., audio-visual sound source localization [44, 12, 61, 4, 33] or audio-visual cross-modal retrieval tasks [57]. Also, when large multi-modal models [18, 49] are extended to multichannel audio, STARSS23 is an appropriate audio-visual soundscape evaluation data for the models because STARSS23 has unique spatial annotations and multichannel audio data beyond stereo.

## 6 Conclusion

This paper attempts to broaden sound event localization and detection (SELD) to an audio-visual area by introducing an audio-visual SELD task. We present an audio-visual dataset, Sony-TAu Realistic Spatial Soundscapes 2023 (STARSS23), which consists of multichannel audio data, video data, and spatiotemporal annotation of sound events in natural sound scenes. Furthermore, we present quantitative evaluations for audio-visual SELD systems compared with audio-only systems and demonstrate the benefits of visual object positions. We still need to improve the SELD performance of various sound events using audio-visual data. Also, STARSS23 opens a wide range of future research on spatial audio-visual tasks, taking advantage of the well-organized audio-visual recording and detailed labels about spatial sound events.

## Acknowledgments and Disclosure of Funding

We thank Akira Takahashi for his helpful code review and thank Atsuo Hiroe, Kazuya Tateishi, Masato Hirano, Takashi Shibuya, Yuji Maeda, and Zhi Zhong for valuable discussions about the data construction process.

The data collection and annotation at Tampere University have been funded by Google. This work was carried out with the support of the Centre for Immersive Visual Technologies (CIVIT) research infrastructure at Tampere University, Finland.

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

# Appendix

We show appendixes for data construction, data analysis, experiment, competition, social impact, and personal data handling. Finally, we answer the questions from *Datasheets for Datasets* [16].

## A    Data Construction

### A.1    Sound Scene Recording

The generic instructions for recording participants include information about duration, participants, props, active classes, and description of sound scenes. The recording duration can be shorter or longer as they are *rough* instructions. We set the basic structure of sound scenes, e.g., people gathering on a sofa and discussing the weekend. Following the rough instructions, we leave the other details of sound scenes to the participants, e.g., they can walk wherever they want and talk about whatever they want without a fixed dialogue text. We show an example of the instructions:

**Duration**  3 min

**Participants**  3 people

**Props**  Playing cards, mobile phone

**Active classes**  Speech, laugh, mobile phone

**Description**

- 3 people gather on the sofa and talk about the weekend.
- Propose to play a game of cards, shuffle, and distribute the cards.
- Mobile phone rings while playing, attend the call, walk around and talk for a few seconds, and laugh during the call.
- Come back, sit, and continue playing.

Sound scenes consist of target sound events, directional interference sounds, and background noise. Each target class contains diverse sounds, e.g., the speech classes include a few different languages, and the phone class has sounds from different mobile phones. In addition, several target classes correspond to super classes with some subclasses in the AudioSet ontology [17], e.g., the domestic sounds class contains the vacuum cleaner, mechanical fan, and boiling subclasses. We provide the subset of sounds encountered in the recordings for the target classes in the form of more specific AudioSet-related labels. The subset information is summarized in Table 5, which is a re-post of a table in the STARSS22 paper [39]. Directional interference sounds are derived from computer keyboards, shuffling cards, dishes, pots, and pans; however, they are not annotated. Natural background noise is mainly related to HVAC (heating, ventilation, and air-conditioning), ranging from low to high levels.

Spatial and temporal annotations relied on carefully mounting tracking markers and wireless microphones [39]. The tracking markers were mounted on independent sound sources such as mobile phones or hoovers. Head markers were also provided to the participants as headbands or hats. The head tracking results were reference points for all human body-related sound classes. Mouth position for speech and laughter classes, feet stepping position for footstep class, and hand position for clapping class were each approximated with a fixed translation from the head tracking result. Regarding clapping, participants were instructed to clap about 20 cm in front of their faces, while footsteps were projected from the head coordinates to floor level. Hence, the mounting positions were considered in the annotation process to translate the tracking data of each class into each sound source position. The wireless microphones were mounted to the lapel of each participant and to additional independent sound sources being far from the participants.

### A.2    Annotation

At a certain time, the motion capture (mocap) system could not track the attached marker, e.g., when markers were out of the view of the mocap coverage, markers were moving too fast, or obstacles occluded markers. Whenever such misses were short, the tracking results were interpolated with Motive[9]. If such misses were long and interpolating the results was not possible, azimuth and

---

[9]`https://optitrack.com/software/motive/`

Table 5: Relation of target classes to specific AudioSet classes. Target classes not included in the table have a one-to-one relationship with the similarly named AudioSet ones.

| Target Class | Related AudioSet subclasses |
|---|---|
| *Telephone* | *Telephone bell ringing*, *Ringtone* (no musical ringtones) |
| *Domestic sounds* | *Vacuum cleaner, Mechanical fan, Boiling* (produced by hoover, air circulator, water boiler) |
| *Door, open or close* | Combination of *Door & Cupboard, open or close* |
| *Music* | *Background music & Pop music*, (played by a loudspeaker in the room) |
| *Musical instrument* | *Acoustic guitar, Marimba, Xylophone, Cowbell, Piano, Rattle (instrument)* |
| *Bell* | Combination of sounds from hotel bell and glass bell, closer to *Bicycle bell* & single *Chime* |

elevation of sound sources were annotated based on the 360° video data. For example, direction of arrival (DOA) of the door class was often annotated from the video data because many doors were outside of the view of the mocap. To calculate source distance in such cases, we also used information on room dimensions and installation positions from our recording log in addition to the video data. Any interpolated spatial annotations were visually checked in the confirmation videos.

### A.3 Data Format

We use WAV file format for the audio data and MP4 file format for the video data. The metadata are tabulated and served in CSV file format. The sound event classes, DOAs, and distances are provided in the following format:

- frame number, active class index, source number index, azimuth, elevation, distance

with all labels served as integers. Frame, class, and source enumeration begin at 0. Frames correspond to a temporal resolution of 100 msec. Azimuth and elevation angles are given in degrees, rounded to the closest integer value, with azimuth and elevation being zero at the front, azimuth $\phi \in [-180°, 180°]$, and elevation $\theta \in [-90°, 90°]$. The azimuth angle increases counter-clockwise ($\phi = 90°$ at the left). Distances are provided in centimeters, also rounded to the closest integer value. The source index is a unique integer for each source in the scene. Note that each unique participant gets assigned one identifier, but not individual events produced by the same participant; e.g., a clapping event and a laughter event produced by the same person have the same identifier. Independent sources that are not participants (e.g., a loudspeaker playing music in the room) get a 0 identifier. Note that the source index and the source distance are only included as additional information that can be exploited during training. An example line could be as follows:

- 10, 1, 1, -50, 30, 181

which describes that in frame 10, an event of class male speech (class 1) belonging to one participant (source 1) is active at location (-50°, 30°, 181 cm).

## B  Data Analysis

### B.1 Duration

Figures 6 illustrates the box plots of duration, i.e., how long a sound event lasts. The figure shows that there are classes with similar duration trends. Speech classes have a wide range of plots, whereas laughing has a shorter duration than speech. While phone and bell have similar medians, the box of phone is longer as phone calls repeat the recorded sound until it is answered. There are several classes with longer duration, e.g., domestic sounds, music, musical instruments, and faucet, whereas collision or tapping sounds such as door and knock are relatively short.

### B.2 DOA

Figure 7 shows the distribution of DOAs of the rest of the classes not depicted in Figure 3. While DOAs of human-produced classes such as speech, laugh, clap, or footsteps are dispersed across the 360° plane, classes such as door, knock, or faucet result in specific discrete points in the plot due to their fixed position in the room. Music and bell classes also show similar trends to door class

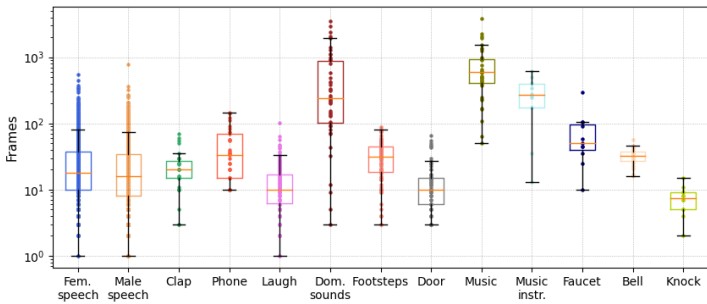

Figure 6: Box plots of the event durations across classes. The vertical axis indicates frame numbers in a log scale. The center line of a box shows the median. The top and bottom lines show the first and third quartile, while the whiskers extend from the box by 1.5 times the interquartile range. Each plot represents a data point of duration.

as their sound sources are rarely moved. Domestic sounds, phone, and musical instrument classes show similar trends to the speech classes as the respective sources are sometimes still and other times moving.

## C  Experiment

### C.1  Evaluation Metric

We describe details on evaluation metrics [25, 38]. The metrics are based on true positives, $TP$, and false positives, $FP$, determined not only by correct or wrong detections but also based on whether they are closer or further than a distance threshold ($20°$ in this paper) from the reference.

We associate $P_c$ predicted events of class $c$ with $R_c$ reference events of class $c$ for each class $c \in [1, ..., C]$ and each segment. We count false negatives for misses: $FN_c = \max(0, R_c - P_c)$, and count false positives for extraneous predictions: $FP_{c,\infty} = \max(0, P_c - R_c)$. $K_c$ predictions are spatially associated with references based on the Hungarian algorithm: $K_c = \min(P_c, R_c)$. The number can also be considered as the unthresholded true positives: $TP_c = K_c$. We apply the spatial threshold to move $L_c(\leq K_c)$ predictions with a value greater than the threshold to false positives: $FP_{c,\geq 20°} = L_c$, and $FP_c = FP_{c,\infty} + FP_{c,\geq 20°}$. We count the remaining matched estimates per class as true positives: $TP_{c,\leq 20°} = K_c - FP_{c,\geq 20°}$.

Based on those, we form the location-aware F-score and error rate. We perform macro-averaging of the location-aware F-score: $F_{20°} = \sum_c F_{20°,c}/C$. Additionally, we evaluate localization accuracy through a class-aware localization error $LE_{CD}$, computed as the mean angular error of the matched true positives per class, and then macro-averaged: $LE_{CD,c} = \sum_k \theta_k/TP_c$ for each segment with $TP_c > 0$, and with $\theta_k$ being the angular error between the $k$-th matched prediction and reference, and after averaging across all frames that have any true positives, $LE_{CD} = \sum_c LE_{CD,c}/C$. When there are no true positive outputs in a class, to compute the macro average, we set $180°$ as the localization error in the class. Complementary to the localization error, we compute a localization recall metric per class, also macro-averaged: $LR_{CD,c} = TP_c/(TP_c + FN_c)$, and $LR_{CD} = \sum_c LR_{CD,c}/C$. The localization error and recall are not thresholded to provide more varied complementary information to the location-dependent F-score, presenting localization accuracy outside the spatial threshold. For a more thorough analysis of the joint SELD metrics, please refer to [25, 38].

### C.2  Experimental Setting

We add a few details on experimental settings. The number of channels on audio features is $M = 7$, consisting of 4ch amplitude spectrograms and 3ch IPDs. We apply the short-term Fourier transform (STFT) to a 1.27-sec audio signal with a 20-ms frame length and 10-ms frame hop. The number of frequency bins is $F = 257$, and the length of time frames is $T = 128$. We keep the input feature length to 128 frames during inference and set the shift length to 120 frames.

For visual features, we use the pre-trained YOLOX object detection model[10], which is trained with the COCO dataset [23]. While COCO has 80 object classes, we focus on person, cell phone, and sink classes because they are strongly related to the 13 sound events in STARSS23. Only person class is stably detected in our preliminary experiments in STARSS23 videos. Therefore, we use the model to get bounding boxes for the person class. We show an example of object detection results in Figure 8. While people are detected by the object detector, a sink is not detected by the detector.

Instead of the current object detection method, we also experimented with two convolutional neural network (CNN)-based visual feature extraction methods, i.e., VGG-16 [47] and ResNet-18 [20]. While training losses of the CNN-based visual features are lower than the object detection method, validation results were worse than the object detection method. Bounding boxes of person class have simple but enough information on human positions. That would lead to generalization on human body-related classes. The CNN-based visual features retain richer visual information than bounding boxes, so they may require more training data to generalize. Visual features extracted from Transformer-based methods, e.g., ViT models [13], would face the same issues as CNN-based methods.

We use a batch size of 16 and the Adam optimizer with a weight decay of $10^{-6}$ to train the audio-visual sound event localization and detection (SELD) system. The learning rate is set to 0.001. We validate and save model weights at every 1,000 iterations up to 20,000 iterations. We select a model that demonstrated the best aggregated SELD error, $\mathcal{E}_{SELD}$, calculated as

$$\mathcal{E}_{SELD} = \frac{ER_{20°} + (1 - F_{20°}) + \frac{LE_{CD}}{180°} + (1 - LR_{CD})}{4}. \tag{6}$$

We report the average scores and error bars of five experiments.

The model parameter size is 0.8 M. The model size is kept intentionally small for easier trials. A single GPU (e.g., GeForce GTX 1080 Ti) is used for training. The training takes around six hours.

### C.3  Experimental Results

In addition to the F-score per class in first-order ambisonics (FOA) format in Figure 5, we show the other SELD metrics per class in both FOA and tetrahedral microphone array (MIC) formats. As shown in Figure 9c, the F-score in MIC format shows a similar trend as in FOA format. While the audio-visual SELD system performs worse in the F-score of non-body-related classes, the audio-visual system demonstrates a higher F-score in body-related classes. Figure 9a and 9d show that the audio-visual system demonstrates lower localization error of body-related classes in both formats. There is no significant trend of the localization recall between the two formats.

## D  Competition

STARSS23 has served as the development set and evaluation set for the SELD Task of the DCASE 2023 Challenge[11], which aims to accelerate audio-only and audio-visual SELD research. The task participants use the development audio/video recordings and labels to train and validate their SELD systems in the development process. The evaluation recordings without labels are used to produce system outputs for the challenge evaluation phase. If researchers wish to compare their system against the submissions of the challenge, they will have directly comparable results if they use the evaluation data as their testing set. Also, the implementation of the audio-visual SELD system described herein, trained and evaluated with STARSS23, has served as the baseline method for the audio-visual track of the challenge.

## E  Social Impact

The STARSS23 enables research on audio-only and audio-visual SELD tasks, which form the backbone of various real-world applications on acoustic and audio-visual monitoring, intelligent home applications, or audio-visual machine perception. The dataset, together with the associated

---

[10]https://github.com/open-mmlab/mmdetection/blob/master/configs/yolox/yolox_tiny_8x8_300e_coco.py

[11]https://dcase.community/challenge2023/task-sound-event-localization-and-detection-evaluated-in-real-spatial-sound-scenes

challenge, accelerates such research for research institutes and industry since it is the first of its kind based on real annotated recordings. Multiple research institutes, university laboratories, and industrial research and development groups have already shown interest in the dataset and its use, either as part of the DCASE challenge or outside of it in independent published studies. We expect the dataset to set the standard in the upcoming years in audio-visual SELD-related studies due to its unique spatial annotations from real tracked people and sound sources and its spatial audio content, which is becoming more and more relevant with multiple monitoring or smart home devices employing microphone arrays. The dataset also offers opportunities for cross-regional evaluation, with its recordings coming from two different sites geographically far apart. Of course, as with many strongly annotated datasets, the diversity of sound events is limited and cannot capture the conditions of many real-world application-specific scenes. However, we believe that it is a valuable contribution to the development and maturation of such systems, at which point we expect more application-specific SELD datasets to appear.

## F    Personal Data Handling

STARSS23 data was recorded with over 50 voluntary participants. Before the recording, we explain our research purpose, how we record the sound scenes, and how we treat and release the recording data. Regarding the recording process, an example of the generic instructions is in Appendix A. Our explanation is based on text format and verbal description. Participants can ask us questions related to recording and public release. We also explain potential risks, i.e., recording data containing personally identifiable information, and our Institutional Review Board (IRB) approvals. The personally identifiable information is raw speech and blurred faces. The participants are also instructed not to reveal personal information during the recordings, and limit themselves to conversations of generic topics. After the explanation and Q&A, when participants understand the purpose and risk of recording and release, each participant signs the consent form.

## G    Data Sheet

For dataset documentation, we take the questions from Datasheets for Datasets [16] and answer them.

### G.1    Motivation

- **Q: For what purpose was the dataset created?** Was there a specific task in mind? Was there a specific gap that needed to be filled? Please provide a description.

  A: This dataset is created to tackle audio-visual sound event localization and detection (SELD) tasks. While visual data about sound source objects can support SELD tasks, e.g., feet are a potential source of footsteps, the existing datasets did not contain the complete multichannel audio, video, and annotation set. STARSS23 serves real sound scene recording with multichannel audio, video data aligned with the audio, and spatiotemporal annotation of sound events. STARSS23 allows the incorporation of audio-visual correspondence into multichannel audio signal processing methods.

- **Q: Who created the dataset (e.g., which team, research group) and on behalf of which entity (e.g., company, institution, organization)?**

  A: Creative AI Laboratory (CAL) at Sony Group and Audio Research Group (ARG) at Tampere University.

- **Q: Who funded the creation of the dataset?** If there is an associated grant, please provide the name of the grantor and the grant name and number.

  A: The data collection and annotation at Tampere University has received funding by Google.

- **Q: Any other comments?**

  A: N/A.

### G.2 Composition

- **Q: What do the instances that comprise the dataset represent (e.g., documents, photos, people, countries)?** Are there multiple types of instances (e.g., movies, users, and ratings; people and interactions between them; nodes and edges)? Please provide a description.

  A: Real sound scene recordings with multichannel audio data, video data aligned with the audio data, and annotations of temporal activation, the direction of arrival (DOA), and distance of sound events.

- **Q: How many instances are there in total (of each type, if appropriate)?**

  A: The development set totals about 7 hours and 22 minutes, of which 168 clips were recorded with 57 participants in 16 rooms with annotation. The evaluation set totals about 3.5 hours and 79 clips without annotation.

- **Q: Does the dataset contain all possible instances or is it a sample (not necessarily random) of instances from a larger set?** If the dataset is a sample, then what is the larger set? Is the sample representative of the larger set (e.g., geographic coverage)? If so, please describe how this representativeness was validated/verified. If it is not representative of the larger set, please describe why not (e.g., to cover a more diverse range of instances, because instances were withheld or unavailable).

  A: It contains all possible instances.

- **Q: What data does each instance consist of?** "Raw" data (e.g., unprocessed text or images) or features? In either case, please provide a description.

  A: Raw audio and video data, but we convert 32ch 48kHz audio recordings to 4ch 24kHz, and 3840×1920 360° video recordings to 1920×960 equirectangular. We also conduct face-blurring on video data to anonymize identical information.

- **Q: Is there a label or target associated with each instance?** If so, please provide a description.

  A: The dataset contains temporal activation, DOA, and source distance labels of 13 target sound event classes. The classes are female speech, male speech, clapping, telephone, laughter, domestic sounds, footsteps, door, music, musical instrument, water tap, bell, and knock.

- **Q: Is any information missing from individual instances?** If so, please provide a description, explaining why this information is missing (e.g., because it was unavailable). This does not include intentionally removed information, but might include, e.g., redacted text.

  A: The evaluation set has no annotation because the set is used in a testing phase of the SELD task of the DCASE 2023 challenge. While almost all audio recordings in the development and evaluation set are accompanied by synchronized video recordings, only 12 audio recordings in the development set are missing videos (from fold3_room21_mix001.wav to fold3_room21_mix012.wav).

- **Q: Are relationships between individual instances made explicit (e.g., users' movie ratings, social network links)?** If so, please describe how these relationships are made explicit.

  A: In each clip, we use the same tag, e.g., foldX_roomY_mixZ. We use the tag for audio (.wav), video (.mp4), and labels (.csv).

- **Q: Are there recommended data splits (e.g., training, development/validation, testing)?** If so, please provide a description of these splits, explaining the rationale behind them.

  A: Yes. STARSS23 has the dev-set-train part and the dev-set-test part for the development process.

- **Q: Are there any errors, sources of noise, or redundancies in the dataset?** If so, please provide a description.

  A: In addition to target sound events, the sound scene recordings contain directional interference sounds and background noise. That makes the dataset a more realistic situation. We confirm all the labels by listening to the audio and watching the video. If there are any errors, they would be negligible.

- **Q: Is the dataset self-contained, or does it link to or otherwise rely on external resources (e.g., websites, tweets, other datasets)?** If it links to or relies on external resources, a) are there guarantees that they will exist, and remain constant, over time; b) are there official archival versions of the complete dataset (i.e., including the external resources as they existed at the time the dataset was created); c) are there any restrictions (e.g., licenses, fees) associated with any of the external resources that might apply to a dataset consumer? Please provide descriptions of all external resources and any restrictions associated with them, as well as links or other access points, as appropriate.

  A: Self-contained.

- **Q: Does the dataset contain data that might be considered confidential (e.g., data that is protected by legal privilege or by doctor–patient confidentiality, data that includes the content of individuals' non-public communications)?** If so, please provide a description.

  A: No.

- **Q: Does the dataset contain data that, if viewed directly, might be offensive, insulting, threatening, or might otherwise cause anxiety?** If so, please describe why.

  A: No.

- **Q: Does the dataset identify any subpopulations (e.g., by age, gender)?** If so, please describe how these subpopulations are identified and provide a description of their respective distributions within the dataset.

  A: STARSS23 set female and male speech as two target sound event classes. The frame coverages of both classes are almost equal: 28.4 % and 31.4 %, respectively.

- **Q: Is it possible to identify individuals (i.e., one or more natural persons), either directly or indirectly (i.e., in combination with other data) from the dataset?** If so, please describe how.

  A: As an audio-visual sound scene recording dataset, STARSS23 contains raw speech data. However, faces in video data are blurred, and the talk contents have no personal topics. So, it is hard to identify individuals. We also explain to participants the potential risk of identical information before recording. After the explanation, participants sign a consent form for recording when they understand the potential risk.

- **Q: Does the dataset contain data that might be considered sensitive in any way (e.g., data that reveals race or ethnic origins, sexual orientations, religious beliefs, political opinions or union memberships, or locations; financial or health data; biometric or genetic data; forms of government identification, such as social security numbers; criminal history)?** If so, please provide a description.

  A: No.

- **Q: Any other comments?**

  A: N/A.

### G.3 Collection Process

- **Q: How was the data associated with each instance acquired?** Was the data directly observable (e.g., raw text, movie ratings), reported by subjects (e.g., survey responses), or indirectly inferred/derived from other data (e.g., part-of-speech tags, model-based guesses for age or language)? If the data was reported by subjects or indirectly inferred/derived from other data, was the data validated/verified? If so, please describe how.

  A: Each clip of a sound scene is recorded in a room with participants and sound props. Participants improvise a scene following a generic instruction about the sound scene and event.

- **Q: What mechanisms or procedures were used to collect the data (e.g., hardware apparatuses or sensors, manual human curation, software programs, software APIs)?** How were these mechanisms or procedures validated?

  A: Multichannel audio and video data are recorded with a 32ch microphone array and 360° camera, respectively. A motion capture system and wireless microphones are also recorded

for annotation. The specific recording equipment is Eigenmike em32[12], Ricoh Theta V[13], Optitrack Flex 13[14], and Røde Wireless Go II[15] respectively.

- **Q: If the dataset is a sample from a larger set, what was the sampling strategy (e.g., deterministic, probabilistic with specific sampling probabilities)?**

  A: N/A.

- **Q: Who was involved in the data collection process (e.g., students, crowdworkers, contractors) and how were they compensated (e.g., how much were crowdworkers paid)?**

  A: Voluntary participants act in a sound scene, and authors record the scene.

- **Q: Over what timeframe was the data collected?** Does this timeframe match the creation timeframe of the data associated with the instances (e.g., recent crawl of old news articles)? If not, please describe the timeframe in which the data associated with the instances was created.

  A: The first round of recordings was collected between September 2021 and April 2022. A second round of recordings was collected between November 2022 and February 2023.

- **Q: Were any ethical review processes conducted (e.g., by an institutional review board)?** If so, please provide a description of these review processes, including the outcomes, as well as a link or other access point to any supporting documentation.

  A: Yes. We get approvals from our Institutional Review Board (IRB). Following the discussion about personally identifiable information, we conduct face-blurring on video data and explain to participants about potential risks, i.e., recording data containing identifiable information.

- **Q: Did you collect the data from the individuals in question directly, or obtain it via third parties or other sources (e.g., websites)?**

  A: From the individuals.

- **Q: Were the individuals in question notified about the data collection?** If so, please describe (or show with screenshots or other information) how notice was provided, and provide a link or other access point to, or otherwise reproduce, the exact language of the notification itself.

  A: Yes. Before the recording, we explain our research purpose, how we record sound scenes, and how we treat and release the recording data. Our explanation is based on text format and verbal description. Participants can ask us questions related to recording and release.

- **Q: Did the individuals in question consent to the collection and use of their data?** If so, please describe (or show with screenshots or other information) how consent was requested and provided, and provide a link or other access point to, or otherwise reproduce, the exact language to which the individuals consented.

  A: Yes. After our explanation and Q&A, when participants understand the purpose and risk of recording and release, each participant signs the consent form.

- **Q: If consent was obtained, were the consenting individuals provided with a mechanism to revoke their consent in the future or for certain uses?** If so, please provide a description, as well as a link or other access point to the mechanism (if appropriate).

  A: Yes. In such a case, they can contact authors.

- **Q: Has an analysis of the potential impact of the dataset and its use on data subjects (e.g., a data protection impact analysis) been conducted?** If so, please provide a description of this analysis, including the outcomes, as well as a link or other access point to any supporting documentation.

  A: N/A.

- **Q: Any other comments?**

  A: N/A.

---

[12] https://mhacoustics.com/products#eigenmike1

[13] https://theta360.com/en/about/theta/v.html

[14] https://optitrack.com/cameras/flex-13/

[15] https://rode.com/en/microphones/wireless/wirelessgoii

## G.4 Preprocessing/Cleaning/Labeling

- **Q: Was any preprocessing/cleaning/labeling of the data done (e.g., discretization or bucketing, tokenization, part-of-speech tagging, SIFT feature extraction, removal of instances, processing of missing values)?** If so, please provide a description. If not, you may skip the remaining questions in this section.

  A: We convert audio and video data to the appropriate size. We also annotate temporal activation and DOA of sound events. After annotating, we confirm the labels by listening to the audio and watching the video.

- **Q: Was the "raw" data saved in addition to the preprocessed/cleaned/labeled data (e.g., to support unanticipated future uses)?** If so, please provide a link or other access point to the "raw" data.

  A: We saved "raw" (large size) audio and video data for future use. If one is interested in them, one can contact the authors.

- **Q: Is the software that was used to preprocess/clean/label the data available?** If so, please provide a link or other access point.

  A: We use Python code for audio data conversion, and RICOH THETA[16] and FFmpeg[17] for video data conversion. To annotate temporal activation, Audacity[18] and REAPER[19] are used. Tracking results are used in Motive[20].

- **Q: Any other comments?**

  A: N/A.

## G.5 Uses

- **Q: Has the dataset been used for any tasks already?** If so, please provide a description.

  A: No, the dataset has not yet been used for any scientific papers. This paper is the first to use the dataset.

- **Q: Is there a repository that links to any or all papers or systems that use the dataset?** If so, please provide a link or other access point.

  A: Yes. We have the code repository of the SELD system: `https://github.com/sony/audio-visual-seld-dcase2023`.

- **Q: What (other) tasks could the dataset be used for?**

  A: Apart from audio-visual SELD tasks, one could use the dataset for audio-only SELD tasks, audio-visual speaker DOA estimation tasks, audio-visual sound source localization tasks, and audio-visual cross-modal retrieval tasks. Audio-visual sound source distance estimation could be another task.

- **Q: Is there anything about the composition of the dataset or the way it was collected and preprocessed/cleaned/labeled that might impact future uses?** For example, is there anything that a dataset consumer might need to know to avoid uses that could result in unfair treatment of individuals or groups (e.g., stereotyping, quality of service issues) or other risks or harms (e.g., legal risks, financial harms)? If so, please provide a description. Is there anything a dataset consumer could do to mitigate these risks or harms?

  A: No.

- **Q: Are there tasks for which the dataset should not be used?** If so, please provide a description.

  A: No.

- **Q: Any other comments?**

  A: N/A.

---

[16] `https://theta360.com/en/about/application/pc.html`

[17] `https://ffmpeg.org/`

[18] `https://www.audacityteam.org/`

[19] `https://www.reaper.fm/`

[20] `https://optitrack.com/software/motive/`

## G.6 Distribution

- **Q: Will the dataset be distributed to third parties outside of the entity (e.g., company, institution, organization) on behalf of which the dataset was created?** If so, please provide a description.

  A: Yes. STARSS23 is publicly available at `https://zenodo.org/record/7880637`.

- **Q: How will the dataset will be distributed (e.g., tarball on website, API, GitHub)?** Does the dataset have a digital object identifier (DOI)?

  A: STARSS23 is distributed via `https://zenodo.org/record/7880637`. The DOI is "10.5281/zenodo.7709051".

- **Q: When will the dataset be distributed?**

  A: STARSS23 is already distributed.

- **Q: Will the dataset be distributed under a copyright or other intellectual property (IP) license, and/or under applicable terms of use (ToU)?** If so, please describe this license and/or ToU, and provide a link or other access point to, or otherwise reproduce, any relevant licensing terms or ToU, as well as any fees associated with these restrictions.

  A: STARSS23 is licensed under MIT License.

- **Q: Have any third parties imposed IP-based or other restrictions on the data associated with the instances?** If so, please describe these restrictions, and provide a link or other access point to, or otherwise reproduce, any relevant licensing terms, as well as any fees associated with these restrictions.

  A: No.

- **Q: Do any export controls or other regulatory restrictions apply to the dataset or to individual instances?** If so, please describe these restrictions, and provide a link or other access point to, or otherwise reproduce, any supporting documentation.

  A: No.

- **Q: Any other comments?**

  A: N/A.

## G.7 Maintenance

- **Q: Who will be supporting/hosting/maintaining the dataset?**

  A: Sony Group and Tampere University.

- **Q: How can the owner/curator/manager of the dataset be contacted (e.g., email address)?**

  A: Please contact `kazuki.shimada@sony.com` and `archontis.politis@tuni.fi`.

- **Q: Is there an erratum?** If so, please provide a link or other access point.

  A: All changes to the dataset will be announced on `https://zenodo.org/record/7880637`.

- **Q: Will the dataset be updated (e.g., to correct labeling errors, add new instances, delete instances)?** If so, please describe how often, by whom, and how updates will be communicated to dataset consumers (e.g., mailing list, GitHub)?

  A: Yes, all the updates will be synced on the website.

- **Q: If the dataset relates to people, are there applicable limits on the retention of the data associated with the instances (e.g., were the individuals in question told that their data would be retained for a fixed period of time and then deleted)?** If so, please describe these limits and explain how they will be enforced.

  A: No.

- **Q: Will older versions of the dataset continue to be supported/hosted/maintained?** If so, please describe how. If not, please describe how its obsolescence will be communicated to dataset consumers.

A: The older dataset versions remain in Zenodo if any changes are made.

- **Q: If others want to extend/augment/build on/contribute to the dataset, is there a mechanism for them to do so?** If so, please provide a description. Will these contributions be validated/verified? If so, please describe how. If not, why not? Is there a process for communicating/distributing these contributions to dataset consumers? If so, please provide a description.

  A: Yes. Others can contact the authors of this paper to describe their proposed extension or contribution. We would discuss their proposed contribution to confirm its validity, and if confirmed, we will release a new version of the dataset on Zenodo and announce it accordingly.

- **Q: Any other comments?**

  A: N/A.

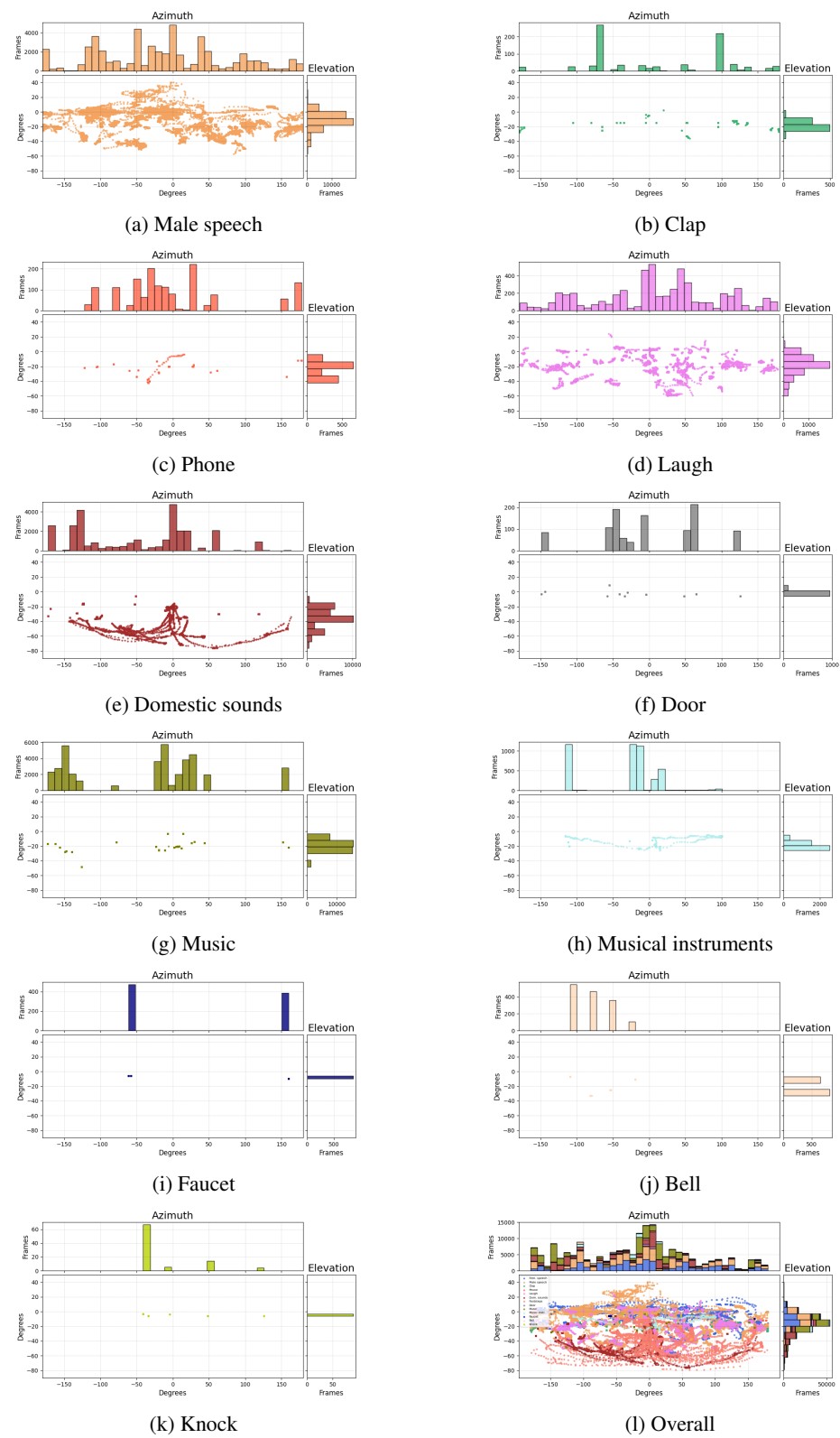

Figure 7: The distribution of azimuth and elevation labels. Each point in the scatter figures represents a frame. The top and right figures are stacked histograms of azimuth and elevation.

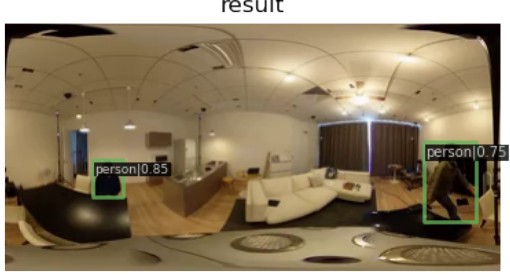

Figure 8: An example of object detection results. While the object detector detects people, the detector does not detect a sink, which is slightly to the right of the center.

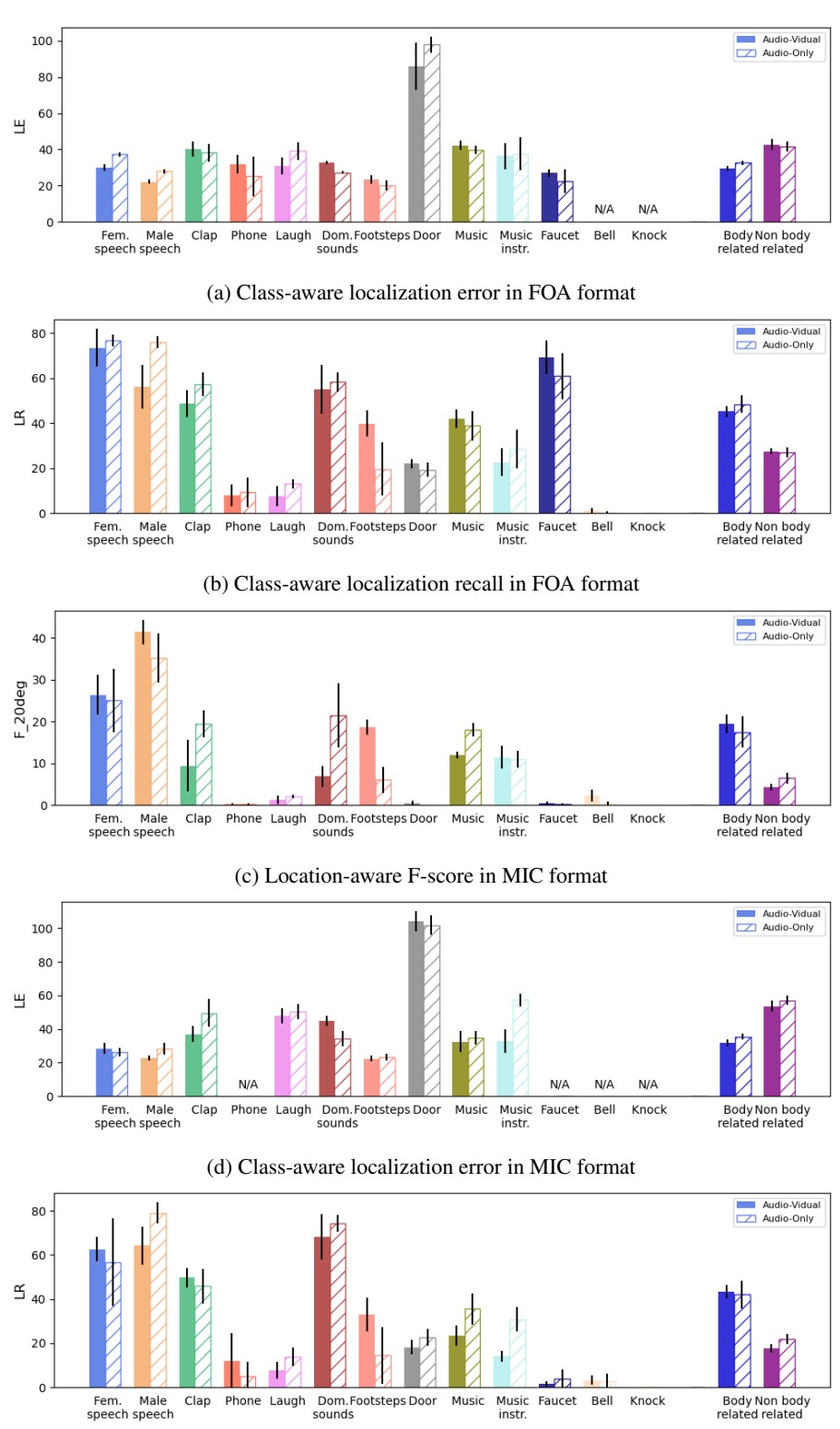

(a) Class-aware localization error in FOA format

(b) Class-aware localization recall in FOA format

(c) Location-aware F-score in MIC format

(d) Class-aware localization error in MIC format

(e) Class-aware localization recall in MIC format

Figure 9: SELD performance ($C = 13$) per sound event class in audio-visual and audio-only systems evaluated for the dev-set-test in STASS23. We found that a few classes sometimes had zero localization recall, i.e., no true positive output in these classes. As statistics of such cases are less meaningful in localization error, we do not show bars and put N/A in the figures.

