# OpenReview forum: "STARSS23: An Audio-Visual Dataset of Spatial Recordings of Real Scenes with Spatiotemporal Annotations of Sound Events"
_NeurIPS.cc/2023/Track/Datasets_and_Benchmarks — NeurIPS 2023 Datasets and Benchmarks Poster_

### Official Review · Reviewer_CWXv · 2023-07-10
**Audio-visual dataset**

**Rating:** 7
**Confidence:** 2
**Correctness:** The authors have correctly claimed th…
**Clarity:** This paper is easy to understand and …

**Strengths:**

13 target sound event classes are annotated. The classes follow loosely the Audioset ontology.

**Additional Feedback:**

Since this work is submitted to the track on Datasets and Benchmarks, it would be beneficial to allocate some space to discuss these aspects.

**Documentation:**

The authors have provided sufficient detail regarding the data collection process, organization, availability, and maintenance.

**Ethics:**

There are no ethical concerns associated with this submission.

**Limitations:**

The authors should discuss more potential tasks such as audio-visual cross-modal retrieval and classification .


**Opportunities For Improvement:**

The research motivation for this work is not clearly stated, and the coverage of related works is insufficient.

**Relation To Prior Work:**

More descriptions are desired.

**Summary And Contributions:**

The authors present a new audio-visual dataset called Sony-TAu Realistic Spatial Soundscapes 2023 (STARSS23). Moreover, they introduced an audio-visual sound event localization and detection (SELD) task that utilizes multichannel audio and video information to estimate the temporal activation and direction of arrival (DOA) of target sound events.

---

> ### Author Response · Authors · 2023-08-22
> **Response to reviewer CWXv**
>
> Dear reviewer,
>
> We thank you for your insightful and positive feedback. We address your comments below and will incorporate your feedback.
>
> **Q1: Research motivation.**
>
> Our research motivation is to investigate how visual feature inputs affect SELD systems.
> The investigation results would improve the performance of applications such as smart-home devices or surveillance systems.
> We will rewrite the introduction to make our motivation more clear.
>
> **Q2: Related works and potential tasks.**
>
> While we mainly introduce STARSS23 for SELD or audio-visual sound source localization tasks, STARSS23 would be applied to other tasks, e.g., audio-visual cross-modal retrieval and classification tasks.
> We will refer to several works such as [a], and discuss how we utilize STARSS23 for audio-visual cross-modal retrieval and classification tasks.
>
> [a] "Deep triplet neural networks with cluster-CCA for audio-visual cross-modal retrieval," Donghuo et al., ACM TOMM, 2020.
>
> **Q3: Aspects of Datasets and Benchmarks.**
>
> The Datasets and Benchmarks track is an appropriate place to discuss improving dataset development.
> We have discussed more general practices in data collection of STARSS23 in the Appendix, e.g., personal data handling practices in Appendix F.
>
> We hope that the clarifications in this reply and the changes to the paper make the content more precise and concise.
>
> Sincerely,
>
> Authors

---

### Official Review · Reviewer_pjsy · 2023-07-22
**Incremental work over STARSS22**

**Rating:** 5
**Confidence:** 5
**Correctness:** The pipeline of dataset construction …
**Clarity:** Clearly written paper.

**Strengths:**

The pipeline of the dataset construction is clear, feasible and reproducible. The baseline setups are also reasonable.

**Additional Feedback:**

NA

**Documentation:**

Good enough.

**Ethics:**

One concern comes from the privacy issues. It is shown that the faces of participants are blurred in the released dataset, however, in practical usage the raw visual information will still be captured. It is not practical to always get permission of all persons that may appear in the camera.

**Limitations:**

The main weakness of the proposed dataset is the limited novelty over the previous work STARSS22. It seems that the work is very incremental in that only visual information is intergrated by adding an additional camera, while other pipelines would remain the same. We understand it can be useful to develop a multimodal dataset for SELD, while the contribution of adding visual information looks to be not novel and significant enough to be considered as a NeurIPS publication.

The experimental results also show that the visual information may not be always useful for SELD as it harms the detection performance (table 3). While it can be claimed that this makes room for algorithm development, the work should at least demonstrate the necessity of using the visual information.

Another concern comes from the privacy issues. It is shown that the faces of participants are blurred in the released dataset, however, in practical usage the raw visual information will still be captured. It is not practical to always get permission of all persons that may appear in the camera.

**Opportunities For Improvement:**

See below.

**Relation To Prior Work:**

Clearly disussed.

**Summary And Contributions:**

The paper develops a new dataset for sound event localization and detection (SELD), which is a well-established task and can be generally tackled by designing a CRNN network, and it is also a conventional task in DCASE challenges. The proposed dataset improves the previous datasets for general sound events by intergrating the visual information, therefore, the SELD problem in the audio-visual case is raised. A CRNN based baseline is also developed and achieves lower localization error than the audio-only systems.

---

> ### Author Response · Authors · 2023-08-22
> **Response to reviewer pjsy (1)**
>
> Dear reviewer,
>
> We thank you for your insightful and positive feedback.
> We address your comments below and will incorporate your feedback.
>
> **Q1: The contribution of adding visual information.**
>
> Integrating audio and visual data in STARSS23 contributes to multichannel audio and video research and progressive multi-modal research.
>
> 1. Multichannel audio and video research:
> There are few annotated datasets with multichannel audio and video, as in Table 1 of the paper.
> Especially our multichannel audio data have four channels, so STARSS23 would stimulate research using video and multichannel audio beyond stereo.
> Spatial annotations on direction-of-arrival (DOA) and distance labels allow us to use STARSS23 as evaluation data of spatial audio-visual tasks, e.g., audio-visual SELD or audio-visual sound source localization tasks.
> While web-curated datasets such as YouTube-360 (Reference [20]) are appropriate for self-supervised training, the unlabeled data is hard to use for evaluation data.
>
> 2. Progressive multi-modal research:
> STARSS23 is not only for the audio-visual SELD dataset but also valuable for other audio-visual tasks, such as audio-visual cross-modal retrieval, as reviewer 7Cu6 suggested.
> Also, recently there has been progress on large multi-modal models, e.g., ImageBind [a] or CoDi [b].
> When such large multi-modal models are extended to multichannel audio, STARSS23 is an appropriate audio-visual soundscape evaluation data for the models because STARSS23 has unique spatial annotations and multichannel audio data beyond stereo.
>
> We will brush up the introduction to make these points more clear.
>
> [a] "ImageBind: One embedding space to bind them all," Girdhar et al., CVPR, 2023.
>
> [b] "Any-to-any generation via composable diffusion," Tang et al., arXiv, 2023.

---

> > ### Author Response · Authors · 2023-08-22
> > **Response to reviewer pjsy (2)**
> >
> > **Q2: The necessity of using visual information.**
> >
> > **Summary**: Our experimental results (Figure 5 in 4.4 and Table 5 in Appendix C.3) show that visual location information of people improves SELD performance in human body-related classes.
> > The results demonstrate the necessity of using visual information.
> >
> > As shown in Table 3, in the average of all 13 classes, visual information improves localization performance, but slightly worsens detection performance.
> > However, Figure 5 shows that the proposed audio-visual systems perform better for human body-related classes, i.e., female and male speeches, clapping, laughing, and footstep.
> > This trend is also seen in additional experiments in Appendix C.3.
> > In the experiments, we set the output of SELD systems to only these five classes.
> > Table 5 in Appendix C.3 shows that the audio-visual systems outperform the audio-only systems on all the metrics when we focus on the human body-related classes.
> > We cite the FOA results below.
> >
> > "System, ER20 $\downarrow$, F20 $\uparrow$, LECD $\downarrow$, LRCD $\uparrow$
> >
> > Audio-Visual, 0.93 $\pm$ 0.08, 22.4 $\pm$ 2.4 \%, 31.0 $\pm$ 3.6 $^{\circ}$, 45.0 $\pm$ 5.0 \%
> >
> > Audio-Only, 0.93 $\pm$ 0.02, 18.0 $\pm$ 1.9 \%, 33.8 $\pm$ 0.8 $^{\circ}$, 40.9 $\pm$ 5.5 \%
> >
> > (from Table 5 in Appendix C.3)"
> >
> > The results of the body-related classes seem natural since current audio-visual systems use bounding boxes only for person class.
> > If visual location information of people is provided, we should expect improvements in the human body-related sound event classes, while we should not expect improvements in the rest.
> > We use only person class because other classes are not stably detected in our preliminary object detection experiments on STARSS23 videos, as described in Appendix C.1.
> > We cite the details below.
> > "For visual features, we use the pre-trained object detection model, which is trained with the COCO dataset. While COCO has 80 object classes, we focus on person, cell phone, and sink classes because they are strongly related to the 13 sound events in STARSS23. Only person class is stably detected in our preliminary experiments in STARSS23 videos. Therefore, we use the model to get bounding boxes for person class. (from Appendix C.1)"
> >
> > We have another supportive result for audio-visual SELD in a competition using STARSS23 (Appendix D).
> > A team has reported that decision-level audio-visual fusion made the audio-visual SELD system better than the audio-only one [a].
> > The decision-level audio-visual fusion method associates the position of some sound event classes with the keypoint detection result of its associated visual classes.
> > For example, the position of footsteps is associated with foot detection results.
> > As the associated sound event classes, they select telephone, water tap, and the five body-related sound event classes.
> >
> > Our experimental results show that visual object detection results of person class improve SELD performance on human body-related classes.
> > The results suggest that the audio-visual SELD systems perform better if visual information matches the target classes of the audio-visual systems.
> > To clarify the point, we will move part of the descriptions in Appendix C.1 and Table 5 in Appendix C.3 and explain that the proposed audio-visual systems perform well in classes related to bounding boxes of person class.
> > In addition, we will describe that there is room for the development of visual features and model structure to improve the performance of other classes.
> >
> > [a] "The NERC-SLIP system for sound event localization and detection of DCASE2023 challenge," Wang et al., DCASE Challenge Technical Report, 2023.
> >
> > **Q3: Privacy issues in practical usage.**
> >
> > As Ethics review jQjq said, the problem that a system captures raw visual information is not unique to audio-visual SELD systems, but also to many other visual machine learning systems.
> > We believe that it is not an issue with this dataset in isolation.
> > We will discuss the privacy issues in the deployment phase of visual machine learning systems, including audio-visual SELD systems.
> >
> > We hope that the clarifications in this reply and the changes to the paper make the content more precise and concise.
> >
> > Sincerely,
> >
> > Authors

---

### Official Review · Reviewer_7Cu6 · 2023-07-24
**An interesting contribution to promote the field of SELD**

**Rating:** 6
**Confidence:** 3

**Strengths:**

The idea of adding another modality input, visual data, to the existing STARSS22 dataset is intuitive. It is good for tackling the task of audio-visual sound event localization and detection. The visual information can benefit temporal activation and DOA, as audible information is usually apparent in the visual modality. This dataset could facilitate research in this field and is therefore an overall positive contribution to the audio, vision, and multimodal research communities.

**Additional Feedback:**

None

**Clarity:**

The paper is well-written and easy to follow, however, there are grammatical mistakes throughout. Given the speakers are non-native, the overall error rate is acceptable, though the paper would benefit from careful proofreading from a native speaker.

**Correctness:**

The paper describes the construction purpose and evaluation clearly, and the idea for construction is novel but reasonable. The multichannel audio data are delivered as two 4-channel spatial formats: FOA and tetrahedral microphone array (MIC). The video data are blurred 1920×960 equirectangular video data recorded by a 360◦ camera. In appendix, author describes the data construction and annotation steps carefully. The labels are manually verified.

**Documentation:**

The data set collection process is described in detail. The logic is correct. Clearly, the fact that STARSS23 is used as SELD task in the DCASE 2023 challenge shows the uses of this dataset.

**Ethics:**

I do not foresee ethical concerns in the way the dataset was curated and distributed. The easy accessibility of STARSS23 is friendly to new beginners.

**Limitations:**

The author mentioned improving SELD performance as their limitation. In my opinion, some analysis of difficult cases can demonstrate in visual would be helpful to the following researchers. Also, the recordings contain a total of 13 target sound event classes, while the total amount of STARSS23 is a little bit scarce. Maybe the authors have plans to expand the dataset in the future.

**Opportunities For Improvement:**

There are not enough experiments of STARSS23 to solve the problem of SELD. To the best of my knowledge, there are plentiful visual feature extraction methods, CNN-based, Transformer-based and so on. It’s better to prove visual information is useful in the task by testing on different models.

**Relation To Prior Work:**

It is clearly discussed how this work is different from previous contributions. Unfortunately, there are not enough comparable experiments. It brings a benchmark to society without a possible solution.

**Summary And Contributions:**

This paper introduces a dataset, STARSS23, for advancing research in audio-visual sound event localization and detection (SELD). Author gathered multichannel audios and videos in natural sound scenes with multiple spatiotemporal annotations. STARSS23 was created in different places, but they follows a strict annotation approach manually to guarantee the uniformity. This is a novel idea, that may have good applications in downstream tasks. The paper also presents an audio-visual SELD system as benchmark, which shows that the system achieves lower localization error than the audio-only one.

---

> ### Author Response · Authors · 2023-08-22
> **Response to reviewer 7Cu6 (1)**
>
> Dear reviewer,
>
> We thank you for your insightful and positive feedback.
> We address your comments below and will incorporate your feedback.
>
> **Q1: Visual feature extraction methods, e.g., CNN-based or Transformer-based.**
>
> Thank you for your suggestion.
> Instead of the current object detection method, we conducted experiments for two CNN-based visual feature extraction methods, i.e., VGG-16 [a] and ResNet-18 [b].
> While training losses of the visual embedding features are lower than the object detection method, validation results were worse than the object detection method.
> Bounding boxes of person class have simple but enough information on human positions.
> That would lead to generalization on human body-related classes.
> The visual embedding features retain richer visual information than bounding boxes, so they may require more training data to generalize.
> Visual embedding features from Transformer-based methods, e.g., ViT models [c], would face the same issues as CNN-based methods.
> We will add the discussion about the other visual feature extraction methods in the method part.
>
> [a] "Very deep convolutional networks for large-scale image recognition," Simonyan et al., arXiv, 2014.
>
> [b] "Deep residual learning for image recognition," He et al., CVPR, 2016.
>
> [c] "An image is worth 16x16 words: Transformers for image recognition at scale", Dosovitskiy et al., arXiv, 2020.
>
> **Q2: To prove visual information is useful.**
>
> **Summary**: Our experimental results (Figure 5 in 4.4 and Table 5 in Appendix C.3) show that visual location information of people is useful in improving SELD performance in human body-related classes.
>
> Figure 5 shows that the proposed audio-visual systems perform better for human body-related classes, i.e., female and male speeches, clapping, laughing, and footstep.
> This trend is also seen in additional experiments in Appendix C.3.
> In the experiments, we set the output of SELD systems to only these five classes.
> Table 5 in Appendix C.3 shows that the audio-visual systems outperform the audio-only systems on all the metrics when we focus on the human body-related classes.
> We cite the FOA results below.
>
> "System, ER20 $\downarrow$, F20 $\uparrow$, LECD $\downarrow$, LRCD $\uparrow$
>
> Audio-Visual, 0.93 $\pm$ 0.08, 22.4 $\pm$ 2.4 \%, 31.0 $\pm$ 3.6 $^{\circ}$, 45.0 $\pm$ 5.0 \%
>
> Audio-Only, 0.93 $\pm$ 0.02, 18.0 $\pm$ 1.9 \%, 33.8 $\pm$ 0.8 $^{\circ}$, 40.9 $\pm$ 5.5 \%
>
> (from Table 5 in Appendix C.3)"
>
> The results of the body-related classes seem natural since current audio-visual systems use bounding boxes only for person class.
> If visual location information of people is provided, we should expect improvements in the human body-related sound event classes, while we should not expect improvements in the rest.
> We use only person class because other classes are not stably detected in our preliminary object detection experiments on STARSS23 videos, as described in Appendix C.1.
> We cite the details below.
> "For visual features, we use the pre-trained object detection model, which is trained with the COCO dataset. While COCO has 80 object classes, we focus on person, cell phone, and sink classes because they are strongly related to the 13 sound events in STARSS23. Only person class is stably detected in our preliminary experiments in STARSS23 videos. Therefore, we use the model to get bounding boxes for person class. (from Appendix C.1)"
>
> We have another supportive result for audio-visual SELD in a competition using STARSS23 (Appendix D).
> A team has reported that decision-level audio-visual fusion made the audio-visual SELD system better than the audio-only one [a].
> The decision-level audio-visual fusion method associates the position of some sound event classes with the keypoint detection result of its associated visual classes.
> For example, the position of footsteps is associated with foot detection results.
> As the associated sound event classes, they select telephone, water tap, and the five body-related sound event classes.
>
> Our experimental results show that visual object detection results of person class improve SELD performance on human body-related classes.
> The results suggest that the audio-visual SELD systems perform better if visual information matches the target classes of the audio-visual systems.
> To clarify the point, we will move part of the descriptions in Appendix C.1 and Table 5 in Appendix C.3 to the main body and explain that the proposed audio-visual systems perform well in classes related to bounding boxes of person class.
> In addition, we will describe that there is room for the development of visual features and model structure to improve the performance of other classes.
>
> [a] "The NERC-SLIP system for sound event localization and detection of DCASE2023 challenge," Wang et al., DCASE Challenge Technical Report, 2023.

---

> > ### Author Response · Authors · 2023-08-22
> > **Response to reviewer 7Cu6 (2)**
> >
> > **Q3: Analysis of difficult cases.**
> >
> > Our experimental results suggest that the audio-visual systems perform well if visual information matches the target classes.
> > In other words, if we cannot get an apparent visual object or feature for a target audio class, it is difficult to improve audio-visual SELD performance for the class.
> > So one of the difficult cases is the object detection failure.
> > For example, the COCO dataset (Reference [16]) has a sink class, but the YOLOX object detector (Reference [11]) cannot detect well in our preliminary experiments.
> > We will show the object detection failure results of the sink class in the Appendix.
> >
> >
> > **Q4: The total amount of STARSS23 is scarce.**
> >
> > As the data collection and annotation of STARSS23 take massive human effort for high-quality recordings and labels, it is not easy to immediately increase data like STARSS23 by dozens of times.
> > While the 7-hour development set of STARSS23 could be sufficient evaluation data for real soundscape, it is relatively small as training data.
> > We have several plans for training data for audio-visual SELD tasks: multichannel audio-visual simulation data, single-modal data, and unlabeled data.
> > As for multichannel audio-visual simulation data, we can use SoundSpaces 2.0 [a], a platform for geometry-based audio rendering for 3D environments.
> > Also, we can use multichannel audio data such as TNSSE datasets (Reference [2,29,28]) to train the audio branch separately.
> > Unlabeled multichannel audio-visual data, e.g., YouTube-360 (Reference [20]), could help the pre-training for audio-visual SELD tasks.
> > We will discuss the plan for training data in the limitation section.
> >
> > [a] "SoundSpaces 2.0: A simulation platform for visual-acoustic learning," Chen et al., NeurIPS DBT, 2022.
> >
> >
> > **Q5: Grammatical mistakes.**
> >
> > Thank you for your suggestion.
> > We will ask native speakers to proofread for grammatical errors.
> >
> > **Q6: Comparable experiments to other works.**
> >
> > The audio-only SELD system in the paper follows previous works such as SELDnet architecture (Reference [1]) and multi-ACCDOA format (Reference [39]).
> > Because the architecture and format are widely used in SELD tasks, the audio-only system is easily comparable with other audio-only SELD works.
> > As our main scope of the paper is the difference between audio-only and audio-visual SELD systems, we omit the comparison among other audio-only SELD systems in the paper.
> > Also, we have organized a competition with STARSS23, as described in Appendix D.
> > That would bring a benchmark to society.
> >
> > We hope that the clarifications in this reply and the changes to the paper make the content more precise and concise.
> >
> > Sincerely,
> >
> > Authors

---

### Official Review · Reviewer_zAcc · 2023-07-28
**The dataset was carefully collected and the research problem is interesting. However, the experiments cannot fully validate the effectiveness of the dataset.**

**Rating:** 5
**Confidence:** 4
**Correctness:** The dataset is constructed in a sound…
**Clarity:** The paper is quite easy to follow.

**Strengths:**

+ The collected Sony-TAu Realistic Spatial Soundscapes 2023 (STARSS23) dataset consists of multichannel audio data recorded with a microphone array, video data, and spatiotemporal annotation of sound events. It enables audio-visual research for spatial audio scene understanding.

+ The proposed audio-visual sound event localization and detection (SELD) task is very interesting. It uses multichannel audio and visual information to estimate the temporal activation and direction of arrival (DOA) of target sound events.

+ The paper is easy to follow. The significance of the dataset is clearly illustrated, and the details about the dataset collection are provided.

**Additional Feedback:**

See Opportunities For Improvement.

**Documentation:**

Sufficient details about the data are provided.

**Limitations:**

Yes.

**Opportunities For Improvement:**

- Dataset imbalance and long-tail. Table 2 shows that the dataset is highly imbalanced and long-tailed. The majority of the data are speech, music, and domestic sounds, while the remaining categories are much less represented. This raises the question of whether models can learn anything meaningful for rare categories.

- The necessity of visual information. The authors claim that the dataset is audio-visual, but the model using both audio and visual information achieves even inferior sound event localization and detection performance. This suggests that visual information is not actually needed for this task. The experimental results do not validate the necessity of the newly collected dataset.

- Metrics. The definitions and formulas of the used metrics should be provided. This would make it easier for other researchers to understand and replicate the results.

- Benchmark section. The benchmark section is not well-written. The used baseline method is not clearly illustrated. I would suggest the authors use symbols and math formulas to organize the proposed method and provide more details.

**Relation To Prior Work:**

The authors clearly discussed the differences with prior work.

**Summary And Contributions:**

The paper proposes an Audio-Visual Sound Event Localization and Detection (SELD) method, which uses both audio and visual data to determine sound event characteristics. An associated dataset, Sony-TAu Realistic Spatial Soundscapes 2023 (STARSS23), includes multichannel audio, video, and spatiotemporal sound event annotations. Benchmark results indicate the SELD system outperforms audio-only systems in localizing sound events.

***Post-rebuttal***

I have read comments from fellow reviewers and the rebuttal. The authors only address my concerns partially. My major concerns are still there. For the dataset imbalance and long tail issue, the authors failed to provide any additional experimental results to justify and address the issue.

I think this is still a very borderline paper. I keep my initial rating unchanged.

---

> ### Author Response · Authors · 2023-08-22
> **Response to reviewer zAcc (1)**
>
> Dear reviewer,
>
> We thank you for your insightful and positive feedback.
> We address your comments below and will incorporate your feedback.
>
> **Q1: Dataset imbalance and long-tail.**
>
> If we simply use STARSS23 training data, the imbalance and long-tail make it difficult for models to learn about rare categories.
> Several machine learning algorithms, e.g., focal loss [a] or label-distribution-aware margin loss [b], tackle the data imbalance problem.
> Regarding the data approach, we can apply sampling strategies [c].
> We also utilize audio-only data of the rare categories [d] for the audio branch training.
> We will discuss the plans to tackle the imbalance problem.
>
> When we consider STARSS23 as an evaluation set, the dataset would be appropriate as it can reflect real soundscape, which is usually imbalanced and long-tailed.
>
> [a] "Focal loss for dense object detection," Lin et al., ICCV, 2017.
>
> [b] "Learning imbalanced datasets with label-distribution-aware margin loss," Cao et al., NeurIPS, 2019.
>
> [c] "SMOTE: synthetic minority over-sampling technique," Chawla et al., JAIR, 2002.
>
> [d] "FSD50k: an open dataset of human-labeled sound events," Fonseca et al., IEEE/ACM TASLP, 2021.
>
> **Q2: The necessity of visual information.**
>
> **Summary**: Our experimental results (Figure 5 in 4.4 and Table 5 in Appendix C.3) show that visual location information of people helps improve SELD performance in human body-related classes.
> The experimental results validate the necessity of the audio-visual dataset.
>
> As shown in Table 3, in the average of all 13 classes, visual information improves localization performance, but slightly worsens detection performance.
> However, Figure 5 shows that the proposed audio-visual systems perform better for human body-related classes, i.e., female and male speeches, clapping, laughing, and footstep.
> This trend is also seen in additional experiments in Appendix C.3.
> In the experiments, we set the target classes of SELD systems to only these five classes.
> Table 5 in Appendix C.3 shows that the audio-visual systems outperform the audio-only systems on all the metrics when we focus on the human body-related classes.
> We cite the FOA results below.
>
> "System, ER20 $\downarrow$, F20 $\uparrow$, LECD $\downarrow$, LRCD $\uparrow$
>
> Audio-Visual, 0.93 $\pm$ 0.08, 22.4 $\pm$ 2.4 \%, 31.0 $\pm$ 3.6 $^{\circ}$, 45.0 $\pm$ 5.0 \%
>
> Audio-Only, 0.93 $\pm$ 0.02, 18.0 $\pm$ 1.9 \%, 33.8 $\pm$ 0.8 $^{\circ}$, 40.9 $\pm$ 5.5 \%
>
> (from Table 5 in Appendix C.3)"
>
> The results of the body-related classes seem natural since current audio-visual systems use bounding boxes only for person class.
> If visual location information of people is provided, we should expect improvements in the human body-related sound event classes, while we should not expect improvements in the rest.
> We use only person class because other classes are not stably detected in our preliminary object detection experiments on STARSS23 videos, as described in Appendix C.1.
> We cite the details below.
> "For visual features, we use the pre-trained object detection model, which is trained with the COCO dataset. While COCO has 80 object classes, we focus on person, cell phone, and sink classes because they are strongly related to the 13 sound events in STARSS23. Only person class is stably detected in our preliminary experiments in STARSS23 videos. Therefore, we use the model to get bounding boxes for person class. (from Appendix C.1)"
>
> We have another supportive result for audio-visual SELD in a competition using STARSS23 (Appendix D).
> A team has reported that decision-level audio-visual fusion made the audio-visual SELD system better than the audio-only one [a].
> The decision-level audio-visual fusion method associates the position of some sound event classes with the keypoint detection result of its associated visual classes.
> For example, the position of footsteps is associated with foot detection results.
> As the associated sound event classes, they select telephone, water tap, and the five body-related sound event classes.
>
> Our experimental results show that visual object detection results of person class improve SELD performance on human body-related classes.
> The results suggest that the audio-visual SELD systems perform better if visual information matches the target classes of the audio-visual systems.
> To clarify the point, we will move part of the descriptions in Appendix C.1 and Table 5 in Appendix C.3 to the main body and explain that the proposed audio-visual systems perform well in classes related to bounding boxes of person class.
> In addition, we will describe that there is room for the development of visual features and model structure to improve the performance of other classes.
>
> [a] "The NERC-SLIP system for sound event localization and detection of DCASE2023 challenge," Wang et al., DCASE Challenge Technical Report, 2023.

---

> > ### Author Response · Authors · 2023-08-22
> > **Response to reviewer zAcc (2)**
> >
> > **Q3: Metrics definitions, formulas, and replications.**
> >
> > Thank you for your suggestion.
> > The paper follows the widely-used metrics for audio-only SELD (Reference [18,31]).
> > We describe the overview of the metrics and refer to the metrics papers in order to make spaces.
> > In addition to the overview, we will add more details to clarify the metrics.
> >
> > Also, we have served the metrics code in our GitHub repository (Note No.8).
> > So it is easy to replicate the metrics.
> >
> > **Q4: Symbols and math formulas in the proposed method.**
> >
> > Thank you for your comments.
> > The proposed method follows the formulas in the SELDnet (Reference [1]) and multi-ACCDOA (Reference [39]).
> > Due to space limitations, we omit the proposed method's symbols and math formulas.
> > We will add the symbols and math formulas with one additional page of camera-ready to provide more details.
> >
> >
> >
> > We hope that the clarifications in this reply and the changes to the paper make the content more precise and concise.
> >
> > Sincerely,
> >
> > Authors

---

### Comment · Area_Chair_CRhw · 2023-08-29
**Please review the rebuttal**

Dear Reviewers,

The author-reviewer discussion period will end in one day. For those reviewers who have not yet review the authors' rebuttals, please do so as soon as possible and consider whether any rating adjustments are necessary. Thank you!

Thanks,
AC

---

### Decision · Program_Chairs · 2023-09-22

**Decision:**

Accept (Poster)

**Comment:**

In general, based on the reviewers' opinions, this work has made some contributions, and the overall rating tends to be positive. Therefore, I believe it is suitable for acceptance at our conference. However, a concern raised by multiple reviewers is the necessity of visual information. Due to the presence of experimental results that could potentially contradict the motivation and theory, the authors should consider providing additional analysis and explanations in the final camera-ready version.